

# Which processes drive observed variations of HCHO columns over India?

Luke Surl[1], Paul I. Palmer[1,2], and Gonzalo González Abad[3]

[1]National Centre for Earth Observation, University of Edinburgh, Edinburgh, UK.
[2]School of GeoSciences, University of Edinburgh, Edinburgh, UK.
[3]Atomic and Molecular Physics Division, Harvard–Smithsonian Center for Astrophysics, Cambridge, Massachusetts, USA

*Correspondence to:* Paul I. Palmer (paul.palmer@ed.ac.uk)

**Abstract.** We interpret HCHO column variations observed by the Ozone Monitoring Instrument (OMI), aboard the NASA Aura satellite, over India during 2014 using the GEOS-Chem atmospheric chemistry and transport model. We use a nested version of the model with a spatial resolution of approximately 25 km. HCHO columns are related to local emissions of volatile organic compounds (VOCs) with a spatial smearing that increases with the VOC lifetime. Over India, HCHO has biogenic, pyrogenic, and anthropogenic VOC sources. Using a 0-D photochemistry model, we find that isoprene has the largest molar yield of HCHO that is typically realized within a few hours. We find that forested regions that neighbours major urban conurbations are exposed to high levels of nitrogen oxides. This results in depleted hydroxyl radical concentrations and a delay in the production of HCHO from isoprene oxidation. We find that propene is the only anthropogenic VOC emitted in major Indian cities that produces HCHO at a comparable (slower) rate to isoprene. The GEOS-Chem model reproduces the broadscale annual mean HCHO column distribution observed by OMI ($r$=0.6), which is dominated by a distinctive meridional gradient in the northern half of the country, and by localized regions of high columns that coincide with forests. Major discrepancies are over the Indo-Gangetic Plain and Delhi. We find that the model has more skill at reproducing observations during winter (JF) and pre-monsoon (MAM) months with Pearson correlations $r$>0.5 but with a positive model bias of $\simeq$1×10^{15} molec/cm$^2$. During the monsoon season (JJAS) we reproduce only a diffuse version of the observed meridional gradient ($r$=0.4). We find that on a continental scale most of the HCHO column seasonal cycle is explained by monthly variations in surface temperature ($r$=0.9), suggesting a role for biogenic VOCs, in agreement with the 0-D and GEOS-Chem model calculations. We also find that the seasonal cycle during 2014 is not significantly different from the 2008-2015 mean seasonal variation but there are large year to year variations. There are two main



loci for biomass burning (states of Punjab and Haryana, and northeastern India), which we find only contributes a significant contribution (up to $1\times10^{15}$ molec/cm$^2$) to observed HCHO columns during March to April over northeastern India. The slow production of HCHO from propene oxidation results in a smeared hotspot over Delhi that we resolve only on an annual mean timescale by using a temporal oversampling method. Using a linear regression model to relate GEOS-Chem isoprene emissions to HCHO columns we infer seasonal isoprene emissions over two key forest regions from the OMI HCHO column data. We find that the *a posteriori* emissions are typically lower than the *a priori* emissions, with a much stronger reduction of emissions during the monsoon season. We find that this reduction in emissions during monsoon months coincides with a large drop in satellite observations of leaf phenology that recovers in post monsoon months. This may signal a forest-scale response to monsoon conditions.

## 1   Introduction

Formaldehyde (HCHO) is an important source of the hydroperoxyl radical (Volkamer et al., 2010; Whalley et al., 2010), and a source of upper tropospheric hydroxyl radical (OH) (Jaeglé et al., 1998). It therefore plays a role in determining the oxidizing capacity of the global troposphere. The principal source of HCHO is from the oxidation of methane (CH$_4$), which provides a global ambient background. Shorter-lived non-methane volatile organic compounds (NMVOCs) elevate HCHO concentrations over continental atmospheres. Minor direct HCHO sources include biomass burning, industry, agriculture, automobiles, shipping, and vegetation. The atmospheric lifetime of HCHO, determined by OH and photolysis, is typically several hours. Building on our past studies (Palmer et al., 2003, 2006; Barkley et al., 2008; Gonzi et al., 2011), we interpret HCHO column distributions over India observed by the Ozone Monitoring Instrument (OMI) aboard the NASA Aura spacecraft (Levelt et al., 2006). We interpret spatial and temporal variations in terms of biogenic, pyrogenic, and anthropogenic VOC sources.

India has the sixth largest economy and the second largest population of any country. It also has one of the largest increases in mortality rates due to chronic exposure to elevated levels of surface ozone and particulate matter (Cohen et al., 2017). Figure 1 shows that India has a rich landscape that includes the Thar desert over northwestern India, major forests over the southwestern coast and over the east and northeast, and five megacities (New Delhi, Mumbai, Kolkata, Bengalura, and Chennai). The Indo-Gangetic Plain (IGP) stretches from eastern Pakistan, across the northern edge of India (bounded by the Himalayas), to Bangladesh. The IGP represents more than a quarter of a million acres of fertile land, which is used primarily to grow rice and wheat, but also maize, sugarcane, and cotton. The southwest monsoon represents the main source of water to the IGP, with contributions also from rivers flowing from the Himalayas. The southwest monsoon begins in June and subsides in September. High temperatures over the Thar desert cause a region of low pressure that helps


to establish a large-scale land-sea breeze with the Indian Ocean. This results in warm, moisture-laden air from the Indian Ocean travelling inland. Eventually, this air meets the Himalayas where it is forced to rise. As the air rises, cooler temperatures result in precipitation. Some areas of India receive ten metres of rain annually, mostly during the monsoon season.

Satellite columns observations of HCHO were originally developed using observed UV spectra from the Global Ozone Monitoring Experiment (Thomas et al., 1998; Chance et al., 2000). HCHO column data are now available from a range of satellite instruments, but here we focus on data from the Ozone Monitoring Instrument aboard the NASA Aura spacecraft. Generally, slant HCHO columns are retrieved by directly fitting to observed spectra in a narrow UV window (Chance et al., 2000; De Smedt et al., 2008; González Abad et al., 2015). Vertical columns are determined by scaling these slant columns by scene-dependent air-mass factors (AMFs), taking into account clouds and aerosol scattering (Palmer et al., 2001). Past work has shown that HCHO columns over India have increased on average between 1.6%/yr (1997–2009, De Smedt et al. (2010)) and 1.5%/yr (1995-2013, Mahajan et al. (2015)). Mahajan et al. (2015) also showed using coincident satellite measurements of HCHO and $NO_2$ that over much of India $O_3$ production is limited by the availability by nitrogen oxides but over urban regions it is limited by the availability of VOCs, supported detailed modelling studies over Delhi (Sharma et al., 2016). Here, we employ a high-resolution ($\simeq$25 km) model of atmospheric chemistry that is closely aligned with the resolution of the satellite data, allowing us to take advantage of the richness of data.

HCHO columns are related to their parent VOC emissions with a smearing spatial scale that is related to the production rate and molar yield of HCHO (Palmer et al., 2003). Past studies have used this approach to infer isoprene emissions from major forests (Palmer et al., 2003; Abbot et al., 2003; Shim et al., 2005; Palmer et al., 2006, 2007; Barkley et al., 2008; Millet et al., 2008; Stavrakou et al., 2009; Curci et al., 2010; Marais et al., 2012; Barkley et al., 2013), biomass burning emissions (Young and Paton-Walsh, 2010; Gonzi et al., 2011; Stavrakou et al., 2016), and anthropogenic emissions (Fu et al., 2007; Stavrakou et al., 2009). Detailed photochemical calculations that link VOCs and the time-dependent production of HCHO lay the groundwork for interpreting the HCHO column data. Many past studies have inferred VOC emissions from the HCHO columns using a linear regression model between these two variables (e.g. Palmer et al. (2003); Millet et al. (2008), but others have adopted a more rigorous Bayesian inverse model approach (e.g. Shim et al. (2005)). Given the large uncertainties associated with VOC emissions (e.g. Guenther et al. (2012)) and the production of HCHO in the low $NO_x$ regime (Wolfe et al., 2016) both approaches provide useful insights. Our study is focused on India where there are significant sources of biogenic, pyrogenic, and anthropogenic VOCs.

The next section describes the OMI HCHO column data, the detailed box-model used to study the time-dependent production of HCHO from VOC oxidation, and the GEOS-Chem atmospheric chemistry transport model focused on India. Section 3 reports the results from our analysis of the



OMI HCHO column data over India, the associated model interpretation of these data, and the iso-prene emissions we infer from the HCHO column data collected over two major forest regions. We conclude in Section 4.

## 2    Data and Methods

Our data are focused on India, as defined by the database of Global Administrative Areas (www.
gadm.org). We adopt climatological definitions of seasons from the Indian Meteorological Depart-ment that are determined by the onset of the regional monsoon system: winter includes January and February; the pre-monsoon season is from March to May; the monsoon season is from June to September; and the post-monsoon season is from October to December. Our focus is on OMI HCHO column variations during 2014, but we put this year into a longer temporal context by comparing it
with data collected from 2008 to 2015.

### 2.1    Ozone Monitoring Instrument HCHO Columns

For our analysis we use HCHO vertical columns from Ozone Monitoring Instrument (OMI), a nadir-viewing UV/Vis spectrometer aboard the NASA Aura satellite that was launched in 2004. Aura is in a sun-synchronous orbit with a local equatorial crossing time of 13:38, achieving daily global
coverage subject to cloud coverage.

OMI uses two imaging grating spectrometers each with a CCD detector to collect solar backscat-tered radiation in the spectral range 270–500 nm using three channels: UV-1 (264–311 nm), UV-2 (307–383 nm), and Vis (349–504 nm). OMI has an across-track swath width of 2,600 km, which in global mode is described by 60 scenes that have ground footprints from $13\times24$ km$^2$ at nadir to
115 $28\times160$ km$^2$ at the swath edges.

Determination of HCHO vertical columns is a two-step procedure, which we describe below. First, slant columns are retrieved from the observed spectra. Second, an air-mass factor (AMF) is used to transform these slant columns into geophysical vertical columns that can be compared more easily with models.

We use the NASA OMHCHOv003 data product (González Abad et al., 2015) from the NASA Data and Information Services Center, which fits HCHO slant columns in the 328.5–356.5 nm window and accounts for competing absorbers, the Ring effect, and undersampling. Typical slant columns range from $4\times10^{15}$ to $6\times10^{16}$ molec/cm$^2$ with associated fitting uncertainties ranging from 30% for larger columns to more than 100% for smaller columns (González Abad et al., 2015).

We adopt a conservative approach to filtering data, based on previous studies (e.g. De Smedt et al. (2015)). We filter slant column data using the main data quality flag (removing suspect, bad or missing scenes); scenes that have been flagged as being affected by the row anomaly corresponding to problem with a row of CCD detectors on OMI; scenes with slant column values greater than





$15 \times 10^{16}$ molec/cm$^2$ that we attribute also to potential row anomalies; scenes with solar zenith angles
less than 0° or greater than 70°; scenes with cloud fractions $> 0.4$; and scenes from the last two rows
at the edge of the swath, which we are too wide to be useful in our analysis. We do not anticipate that
these data filters will introduce bias in our anlysis, with the exception of removing cloudy scenes that
introduces a clear-sky bias. To evaluate the clear-sky bias, we compared the corresponding model
HCHO columns (described below) with and without these cloudy scenes and find that removing
them results in a monthly mean positive bias of 13%, consistent with previous work (Palmer et al.,
2001).

To transform each observed slant column to a vertical column we calculate an AMF that accounts
for temporal and spatial variations from scattering due to clouds and aerosols, and for the vertical
distribution of HCHO. This approach is described in detailed by Palmer et al. (2001); Martin et al.
(2003), and has since been evaluated in a large number of studies (e.g., Palmer et al. (2003, 2006);
Fu et al. (2007); Millet et al. (2008); Barkley et al. (2008); Curci et al. (2010); Gonzi et al. (2011);
Marais et al. (2012)). We use the OMI OMCL02 cloud data product, and the nested GEOS-Chem
model to provide information about aerosols and HCHO vertical distributions, described below.

Finally, as a post-processing step, we remove any large-scale biases using a reference sector nor-
malization procedure (Palmer et al., 2003; Martin et al., 2003). Over the remote Pacific we expect
observed HCHO columns to be determined exclusively by the oxidation of methane. By anchoring
observed values over the remote Pacific (0–40°N, 160–180°W) to the corresponding GEOS-Chem
model values we determine the monthly bias that we subsequently subtract from the data within the
study domain.

## 2.2 Models of Tropospheric Chemistry

We use the CAABA box model, including a comprehensive description of atmospheric chemistry,
to understand the time-dependent yield of HCHO from the oxidation of VOCs in Indian forest and
urban environments. These calculations help determine which VOC emissions are responsible for
observed HCHO column variations. We also use a nested version of the GEOS-Chem 3-D atmo-
spheric chemistry model to interpret the satellite data and to understand chemistry on a regional
spatial scale. We use this model to establish a relationship between emissions of VOCs and HCHO
columns that is used to infer emissions of VOCs that correspond to observed HCHO columns.

### Box modelling

We use v3.0 of the CAABA/MECCA (0-D) box model (Sander et al., 2011) to estimate the time-
dependent production of HCHO from the chemical oxidation of different VOCs in forest and urban
photochemical environments (Table 1).

We set up the model to describe a well-mixed summertime boundary layer, with photochemistry
driven by a diurnal cycle in sunlight. For each environment, the photolysis model JVAL Sander et al.



(2014) is used to calculate photolysis rates assuming clear-sky conditions given latitude, longitude,
and day of year. For each study location, we assume constant values for humidity, pressure, temperature and boundary layer height, taken from colocated GEOS-FP meteorology (see below). For forested environments we also prescribe fixed mixing ratios of $O_3$, $NO_2$ and CO from the GEOS-Chem model, described below. For the Delhi urban environment we use daytime average surface air pollutant values in 2014 reported by Tyagi et al. (2016): $O_3$ (37 ppbv), CO (2.3 ppmv), and $NO_2$
(19 ppbv), Table 1. Figure 1 shows the three forested regions (denoted by E, NE, and S) we chose to explore the range of photochemical environments. For each calculation, we spun-up the model from initial conditions for 48 hours before running the model for a further seven days. We use a model timestep of two minutes.

   To evaluate the time-dependent HCHO yield of anthropogenic and biogenic VOCs we run paired
calculations: 1) a control run and 2) a run in which we perturb a VOC by approximately $5 \times 10^{14}$ molec $cm^{-2}$ over a 15 minute period at 0900 local time on the first day after the two-day spin-up period. The perturbed amount is sufficiently small that we can assume that the chemistry response is approximately linear so that we can compare the perturbed run with the control run. We can then use control minus perturbed model calculations to determine HCHO per-carbon yield from the oxidation
of the perturbed VOC.

**GEOS-Chem 3-D Modelling**

We use v10-01 of the GEOS-Chem global model of atmospheric chemistry and transport (www.geos-chem.org), driven by GEOS-FP analyzed meteorological fields, provided by the Global Modeling and Assimilation Office (GMAO) at NASA Goddard Space Flight Center. The native spatial
resolution of these data is $0.25°$ (latitude) $\times 0.3125°$ (longitude), and includes 47 vertical terrain-following signal-levels that describe the atmosphere from the surface to 0.01 hPa of which 30 are in the troposphere. The 3-D meteorological data are updated hourly, and 2-D fields and surface fields are updated every 3 hours.

   We use the nesting capability of the model to focus on India, defined here as 0–40°N, 65–100°E,
using the native resolution of the meteorological data. We use time-dependent lateral boundary conditions archived from a self-consistent, $4° \times 5°$ version of the global full-chemistry model, which is initialized in January 2013 to minimize the influence of initial conditions.

   Monthly anthropogenic emissions for India are taken from the MIX inventory (Li et al., 2017), including NO, CO, $SO_2$ and NMVOCs that are available on a spatial resolution of of $0.25 \times 0.25°$. The
emissions for India are a mosaic of regional emission inventories, including the Regional Emission inventory in ASia (REAS) (Kurokawa et al., 2013) and those developed by the Argonne National Laboratory (Lu et al., 2011; Lu and Streets, 2012). Indian NMVOC emissions represent by mass 17 Tg per year, of which 43% is from the residential sector, 36% is from transport, 20% is from industry, and a small amount is attributed to the power sector.



Approximately 20% of India's geographical area is described as forest. To describe biogenic VOC
emissions of isoprene, monoterpenes, alkenes, and acetone we use the MEGAN emission model
(Model of Emissions of Gases and Aerosols from Nature, Guenther et al. (2012)). Figure 2 shows
seasonal isoprene emissions over India. Monoterpenes emission have a similar distribution to iso-
prene but are a factor of five smaller in magnitude. Given their relatively small influence on HCHO
column variability (Palmer et al., 2003) we do not discuss them further. Within GEOS-Chem, we
adjust the MEGAN model with time-dependent local environment conditions (e.g., surface temper-
ature, photosynthetic active radiation, and leaf area index, LAI).

Monthly pyrogenic emissions are taken from the Global Fire Emissions Database (GFED) version
4 (van der Werf et al., 2017), which are available with a spatial resolution of $0.25 \times 0.25°$. For a
comprehensive description of the emissions inventories used by GEOS-Chem the reader is referred
to the GEOS-Chem website (http://geos-chem.org).

To interpret OMI HCHO column data we sample the model at the local time averaged between
1300 and 1500, corresponding to the overpass time of OMI, and the location of each observed
scene. To put 2014 into a broader temporal context, we interpreted OMI HCHO columns for years
2008–2015. Due to computational limitations, we used the 2014 model run for the other years,
ignoring year-to-year changes in atmospheric transport. We find that the AMF plays only a minor
role in determining OMI vertical column variations so we do not expect our approach to significantly
influence our study of year to changes in the HCHO columns.

## 3   Results

First, we report results from the box-model calculation to provide some insights into the VOCs that
determine the observed HCHO column variations. We then report the annual and seasonal spatial dis-
tributions of OMI and GEOS-Chem HCHO columns. Using correlative space-borne data we explore
the role of biogenic, pyrogenic, and anthropogenic emissions in determining the spatial distributions
of HCHO. Finally, using a model relationship between isoprene emissions and HCHO columns we
infer isoprene emissions that are consistent with the OMI vertical columns.

### 3.1   HCHO Yield from VOC Oxidation in Urban and Forest Environments

We use the CAABA/MECCA box model to determine the time-dependent production of HCHO
from the VOCs that we expect to be emitted from urban and forest environments over India. This
represents important groundwork for interpreting the satellite observations of HCHO. Table 1 pro-
vides an overview of the box model calculations we describe below.

Relating HCHO column variations to emissions of its parent VOC requires that the VOC 1) has
a high yield of HCHO so that eventual concentrations are elevated above the global background,
determined mainly by the oxidation of $CH_4$; and 2) produces HCHO rapidly so that most of the





HCHO is produced close to the emission source and not smeared over long spatial scales. Isoprene
is the dominant HCHO precursor over many northern midlatitude and tropical forest ecosystems
(Palmer et al., 2003, 2006; Barkley et al., 2008; Curci et al., 2010). Over tropical latitudes biomass
burning emissions of VOCs also play a role in HCHO column variations (Fu et al., 2007; Barkley
et al., 2008; Gonzi et al., 2011; Marais et al., 2012). Fu et al. (2007) also showed that reactive
anthropogenic VOCs played a role in HCHO column variations over China.

**Biogenic VOCs from Forest Environments**

We explore three contrasting forest regions throughout 2014 (Figure 1), characterized by latitude-
dependent levels of photosynthetically active radiation (PAR) and by their proximity to urban emis-
sions. We focus on the yield of HCHO from the emission of isoprene (Palmer et al., 2003).

For each calculation we report in Table 2 the resulting lifetime of the injected isoprene (section
2.2), and the cumulative per C HCHO yield. We find that the duration of the calculation is sufficiently
long that the peak HCHO had time to diminish to a negligible amount.

The atmospheric lifetime of isoprene against OH is typically much shorter than an hour, with
shortest values (<15 mins) during summer months. The most southern site in the state of Kerala
(Figure 1) is where isoprene has the longest lifetime. This is due to OH being suppressed by high
ambient $NO_2$ concentrations (in excess of 8 ppbv) originating from the coastal conurbation in that
state. We find that levels of $NO_x$ are sufficiently high (>1 ppbv) in all forested regions such that
isoprene peroxy radicals preferentially react with NO to rapidly produce HCHO. Peak values for
HCHO production are typically reached within 90 minutes of the isoprene getting oxidized. The OH
suppression in Kerala is primarily responsible for the slow production of HCHO. We find that the
peak HCHO signal is typically reached within 30-90 minutes, corresponding to a smearing length
scale of 9–27 km assuming an example wind speed of 5 m/s. For the Kerala region where the time
taken to reach the peak HCHO signal is typically 2–3 hours the corresponding smearing length
scale is 36–54 km. Even in Kerala the smearing length is comparable to the longitudinal extent of a
single OMI scene and well within the swath width (section 2). Figure 3 shows that the corresponding
values of the per-C HCHO yield ranges from 0.38 to 0.66. Higher values are generally associated
with higher values of $NO_x$, consistent with previous studies (Palmer et al., 2006; Barkley et al.,
2013).

**Anthropogenic VOC in urban environment**

To study the role of anthropogenic non-methane VOCs (NMVOCs) in determining HCHO columns
we simulate boundary layer chemistry over Delhi using emissions from the MIX emission inventory
(Li et al., 2017). Over major cities, NMVOC emissions originate mainly from stationary combus-
tion and the transport sector. The species in the MIX inventory are defined following the SAPRC-
99 chemical mechanism (Carter, 2000). We find that over Delhi, according to the MIX inventory,





NMVOC emissions include propane, propene and ALK4, representing 13%, 32%, and 55% of
NMVOC C emissions. ALK4 denotes $>C_4$ alkanes. The longest chain alkane in the 0-D chemi-
cal mechanism is n-butane. We denote this urban model scenario as "D05" (Table 1). We calculate
individual HCHO yields from the oxidation of propane, propene, and n-butane.

Propene ($CH_3CH=CH_2$) has an atmospheric lifetime of several hours, determined by OH addition
to its double bond. The major intermediate products include HCHO and acetaldehyde. The ultimate
HCHO yield from the oxidation of propene (close to 0.5) is comparable with the value from isoprene
oxidation (Figure 3) but it only reaches 50% of that final value within 12 hours, consistent with the
longer lifetime of propene. Propane and n-butane have atmospheric lifetimes of 10 and 5 days,
respectively, determined by OH. Long-lived intermediate oxidation products include acetone that
further delay the production of HCHO (Figure 3).

Assuming an average wind speed of 3 m/s, taken from wind measurements at Indira Ghandi airport
in Delhi, only HCHO production from the oxidation of propene (out of all the gases that represent a
major contribution to the regional emission inventory) would produce a signal that could potentially
be distinguishable above the ambient concentration. We show later there is some evidence that we
can observe a Delhi HCHO hotspot but only after temporally oversampling the data.

**3.2  OMI and GEOS-Chem Model HCHO Column Distributions**

**Data Filtering and AMF Statistics**

Using the data quality criteria, as defined above, we remove 58% of all OMI HCHO measurements
collected during 2014. The proportion of data removed per month varies during the year, with most
scenes removed during the cloudy monsoon season in July (71%) and the least number of scenes
removed during March (45%).

Figure 4 shows the annual mean distribution of AMFs across India. AMF values range from 0.7
to 1.5 with a median value of 1. AMF values are highest in the Himalayas at the extreme North
of India, which are often covered by snow, and over the Rann of Kutch salt marsh. In both cases,
elevated AMF values are due to high surface albedos.

To determine the influence of the AMFs on the spatial distribution of HCHO vertical columns
we compare fitted OMI slant HCHO columns and the corresponding vertical columns to the GEOS-
Chem model. We find that the vertical columns reproduce 11% more of the spatial distribution,
consistent with past studies (Palmer et al., 2001; Millet et al., 2008) that show that the AMF plays
only a minor role in the observed spatial distribution of vertical HCHO columns.

To investigate the role of model resolution in our calculation of vertical columns we repeated the
annual mean analysis using the $2° \times 2.5°$ version of the GEOS-Chem driven by the same inventories
as described above. Figure 5 shows that the coarser resolution model fails to reproduce smaller-
scale variations, as expected, that define much of the India western coast, and misses variations over





northeastern India. While the model via the AMF calculation only contributes a small amount to the
distribution of HCHO vertical columns it does affect the magnitude of the columns. We find that
there is an overwhelming argument to justify the use of the higher-resolution model.

**Annual Mean Spatial Distribution**

Figure 5 shows the annual mean model and observed HCHO columns for 2014. The lowest observed
and model columns are in the north of the study region coinciding with the Himalayas and (to a lesser
extent) in the Thar desert in the Northwest. The observations show three main regions associated
with elevated HCHO columns: 1) the non-Himalayan part of the Northeast region of India, east of
Bangladesh; 2) territory at the south of the country, around the southern part of the Western Ghats
mountain range, roughly following the borders of the state of Kerala; and 3) a broad area over the
east of the country, roughly outlined by the states of Chattisgarh, Odisha, and the northern part of
Anhdra Pradesh.

The nested GEOS-Chem model reproduces the observed magnitude and broad-scale spatial dis-
tribution of OMI HCHO columns over India (Figure 5), but is much smoother. The model has a
small but positive bias of 14% (12%) for the mean (median) annual column amounts. We find that
the model captures 33% ($r$=0.58) of the observed annual mean spatial variation of HCHO columns.
The major discrepancy between the model and observed annual mean HCHO distributions is over
the IGP and over Delhi. Observed HCHO columns are elevated over the IGP but not to the values
shown in the model, which we discuss below. A HCHO hotspot over Delhi is apparently absent from
the OMI columns, but as we show later there is evidence that the hotspot exists in the observations.

Based on the box modelling, the largest source of HCHO column variability is expected to be
isoprene emissions. Figure 1 shows that the broad-scale annual distribution of HCHO over India is
consistent with the annual mean distribution of Normalized Diffusive Vegetation Index (NDVI) with
a Pearson correlation of $r = 0.49$. Below, we use seasonal variations of HCHO columns to improve
understanding of the drivers.

### 3.3 Seasonal Spatial Distributions

Figure 6 shows OMI and GEOS-Chem model HCHO columns for seasons during 2014. There are
distinct seasonal cycles to the HCHO columns over three broad regions: northeastern and south-
western India, and over the IGP. Over (north)eastern India HCHO columns peak in pre-monsoon
and monsoon months. Columns over southwestern coast of India peak in winter and pre-monsoon
months and are very low in other seasons. Columns over the IGP are elevated above those elsewhere
in northern Indian during pre-monsoon months and peak in monsoon months. Figure 7 shows that
the timing of these elevated columns coincide with warmer surface temperatures. This suggests a
role for biogenic VOC emissions, consistent with the strong, local relationship between isoprene
emissions and HCHO production that we demonstrated above.



The GEOS-Chem model captures these broad-scale observed distributions of HCHO but is much smoother, as expected. We find the model has some skill at reproducing the observed spatial distribution with a Pearson correlation $r$ of $>0.4$, with larger values in winter and pre-monsoon months ($r=0.50$ and $0.62$, respectively) and smaller values in monsoon and post-monsoon months ($r=0.39$ and $0.44$, respectively). On an annual timescale the model has a positive bias of $1.0\times10^{15}$ molec/cm$^2$ (33%), which is skewed due to a large bias during the monsoon months ($1.9\times10^{15}$ molec/cm$^2$, 17%). The model has a more defined peak over Delhi, peaking in monsoon months, but there is only a small, diffuse peak in the observations. We discuss this discrepency below.

Figure 7 shows that on a continental scale the seasonal distribution of HCHO columns over India in 2014 is not significantly different from those from 2008 to 2015. In general, HCHO columns follow a similar cycle in the winter and pre-monsoon months, but there are substantial year to year variations during the monsoon and post-monsoon months suggesting a role for large-scale variation in meteorology associated with the monsoon. Figure 8 shows that the corresponding differences in the spatial distribution of HCHO columns of 2014 compared to the mean 2008–2015 distribution. Median HCHO columns for 2014 are slightly higher than the eight-year average by $0.4\times10^{15}$ molec/cm$^2$, but show a similar spatial distribution taking into account year to year variability (Figure 7).

**Biogenic VOCs**

Figure 7 shows that the observed continental-scale HCHO seasonal cycle is reproduced by the model but with a positive model bias typically within $1\times10^{15}$ molec/cm$^2$. On this spatial scale, most of the seasonal cycle can be explained by variations in surface temperature, suggesting a larger than expected role for biogenic emissions. However, the model fails to capture the elevated monthly column during October 2014. We find this is driven mostly by observed variations in HCHO columns over tropical India, coinciding with the withdrawal of the monsoon from India between late September and mid-October (Pai and Bhan, 2015). We suggest this variation represents a response from the vegetation that is missing in the model. Model error associated with cloud coverage and consequent errors associated with the partioning between direct and diffuse PAR could result in large-scale changes in biogenic emissions.

The only forested region that is the exception to the continental-scale picture is over Kerala (denoted by S in Figure 1). Kerala has a tropical maritime climate with little seasonal variation in temperature. The forested region in Kerala neighbours an urban conurbation associated with a high level of NO$_x$ (Figure 9), which influence the HCHO yields from the oxidation of biogenic VOCs, as described above. Figure 10 shows that the observed seasonal distribution of HCHO columns at this site, broadly reproduced by the model, peaks during the winter and is lowest during pre-monsoon and monsoon months. The size of the seasonal variation in HCHO columns is not fully explained by the small seasonal variation in surface temperature, suggesting a role for an additional driver. Figure 10 shows that over this region, the seasonal distribution is driven mostly by changes in leaf



phenology rather than temperature or PAR. Satellite observations of LAI drop from $\simeq 4$ m$^2$/m$^2$ to
$\simeq 1$ m$^2$/m$^2$ during the monsoon months and recover in the post-monsoon months. We find similar behaviour over the east and northeastern forests with large reductions in LAI during monsoon months (not shown). This behaviour is consistent with vegetation taking advantage of the decreased temperatures and higher precipitation rates during the monsoon season to regulate leaf flushing.

Past work has found a contrasting relationship between leaf phenology and satellite observations of HCHO columns over the Amazon basin (Barkley et al., 2009) in which HCHO columns and LAI values dropped in the transition period between the wet and dry seasons, and recovered soon afterwards. Leaves produce isoprene as a thermotolerance mechanism (Singaas et al., 1997), with new leaves having a higher capacity to produce isoprene that deteriorates with age. To explain the

variation in HCHO columns, Barkley et al. (2009) proposed widescale leaf flushing that allowed vegetation to maximize their protection against the light-rich environment of the dry season. A demographic model of leaf phenology based on the hypothesis that trees seek an optimal LAI as a function of available light and soil water (Caldararu et al., 2012, 2014) explained the observed increase in LAI over the Amazon basin during the dry season as a net addition of leaves in response to

increased solar radiation.

**Pyrogenic VOCs**

The main loci for biomass burning are: 1) a region approximately encompassed by the state boundaries of Punjab and Haryana (Figure 1) and 2) Northeastern India. The states of Punjab and Haryana have two growing seasons: May–September (rice) and November–April (wheat). Paddy stubble

burning in May and October/November represents agriculture burning of wheat and rice residue, respectively. For our purposes Northeastern India includes the Seven Sister states (Arunachal Pradesh, Assam, Meghalaya, Manipur, Mizoram, Nagaland, and Tripura), where there are significant forest fires particularly during March and April. This is mainly due to deforestation to convert forests to agricultural land (Santenda and Kaushik, 2014). These two geographical regions account for more

than 70% of these emissions during 2014.

To illustrate the impact of fires on HCHO column variations we use MODIS firecount data (Justice et al., 2002) to identify when and where fires occur over the states of Punjab and Haryana, and Northeastern India. We calculate monthly mean HCHO vertical columns and firecounts. We then determine which $0.25° \times 0.3125°$ grid cells are most affected by fires by selecting thes cells in the

top 20th percentile of cumulative fires.

Figure 11 shows that the highest number of firecounts generally correspond to when there is the largest difference between all grid cells and those most affected by fire. This difference is relatively small over the states of Punjab and Haryana with a peak value of $1.5 \times 10^{15}$ molec/cm$^2$ during May. There is a large difference during March and April over the Northeastern India where fires contribute


up to $5 \times 10^{15}$ molec/cm$^2$ to the monthly mean. We find that this contribution to HCHO columns is
localized in time and geography.

**Anthropogenic Hotspots**

Guided by *a priori* emissions and our box modelling, we anticipate that propene is the only an-
thropogenic VOC likely to produce HCHO sufficiently rapidly that we can relate elevated HCHO

columns to emissions. However, we find little evidence that seasonally-averaged OMI HCHO columns
are elevated over Indian megacities due to limits in the signal to noise, in agreement with previous
work (Mahajan et al., 2015).

We use a temporal oversampling approach, following Zhu et al. (2014), to improve the spatial
resolution of HCHO columns over Delhi and the surrounding region. Oversampling increases the

signal-to-noise ratio and allows for inspection of finer spatial features, at the expense of the tempo-
ral information. We focus on Delhi because the National Capital Region has population of approxi-
mately 17 million people (Perianayagam and Goli, 2012) over a geographical area of approximately
2000 km$^2$. Based on the MIX emissions inventory, which is indicative of values from 2010, (Figure
6) we expect to see an elevated signal from this region. This bottom-up emission inventory like over-

estimates emissions from the transport sector, which has seen the biggest change from 2010 to 2014
(per. comm.: Jun-ichi Kurokawa, Japan Environmental Sanitation Center, October 2017).

First, our area of focus is divided into a very high resolution grid ($0.02° \times 0.02°$). The temporally-
averaged column for each point in this grid is the average of the OMI observational vertical columns
colected during 2014 with the centre point within 43 km in both the latitudinal and longitudinal

directions. This effectively smears out these observations over 43 km squares. Here, we average
over $43 \times 43$ km$^2$ squares rather than the 24 km radius circles), sampling all the 2014 vertical HCHO
columns from the observational dataset for the area around Delhi.

Figure 12 shows that the oversampling method results in distinct elevated HCHO columns over
New Delhi and along major roadways, although the gradients are still noisy. The magnitude of this

elevation is a $\simeq 10^{15}$ molec/cm$^2$. Elevated areas to the East and Southeast of the city may represent
HCHO produced from VOC transported downwind from Delhi. Based on our results we find that
anthropogenic emissions do not appear to play a large role in the observed column variations of
HCHO over India.

However, our use of HCHO columns from the OMI instrument, which has a local overpass time

of 1330, may be hindering our ability to observe the anthropogenic contribution to HCHO. Biogenic
emissions and to a lesser extent biomass burning emissions peak in early afternoon hours, which is
ideal for OMI. The transportation sector, a major source of anthropogenic VOCs, has peaks during
early morning and late afternoon associated with commuter traffic. We argue that the early morning
0930 overpass of the Global Ozone Monitoring Experiment (GOME-2) aboard the MetOp satellite is

better suited to capture these anthropogenic emissions. Secondary production of HCHO is generally





larger than direct emissions of HCHO, and will occur a few hours after the peak commuter time (e.g. Lin et al. (2012); Wang et al. (2017)). The early morning rush hour in Delhi starts after 0700 so we expect a 0930 overpass to also capture some fraction of the secondary HCHO production. Using data collected using morning and afternoon overpass times to describe diurnal variations of HCHO

was presented by De Smedt et al. (2015), but they did not discuss the relative importance of different VOC emission sources at these different times.

Figure 13 shows the annual mean HCHO columns observed by GOME-2 and OMI. For our preliminary argument we are interested only in the distribution of HCHO columns. Here, we have standardized these data so that that they have a mean of zero and a unit standard deviation using:

$$z_i = \frac{x_i - \bar{x}}{s},\tag{1}$$

where $x_i$ is a data point, $\bar{x}$ is the sample mean, and $s$ is the sample standard deviation. This allows us to compare the two data without worrying about bias. The resulting z-scores represent the number of standard deviations from the population mean. We find that GOME-2 data has higher columns over the IGP, while OMI more clearly emphasizes the forested regions that we can identify indepen-

dently through LAI or NDVI measurements (Figure 1). This qualitative test appears to support our hypothesis but needs further work outside the scope of this current study. Taking advantage of the complementary information from multiple sensors that have different local overpass time requires a sophisticated inverse model approach.

### 3.4   Inferring Isoprene Emissions from OMI HCHO Columns

Based on our analysis of the HCHO yields from Indian VOC sources, and the distribution of observed HCHO columns we conclude that the majority of the observed HCHO column variation is due to biogenic VOC emissions. Here, we adopt a simple inversion methodology based on linear regression to infer isoprene emissions from OMI HCHO columns (Palmer et al., 2006).

First, we filter HCHO column data over India to minimize any interference from pyrogenic and an-

thropogenic sources. We focus on two relatively remote areas of India: "East" defined approximately as the area spanning 16–25°Nand 76–87°N, and "Northeast" defined as the region of India east of 90°E(Figure 1). We remove scenes that correspond to MODIS land cover classifications (Friedl et al., 2010) corresponding to croplands (including cropland mosaics), urban/built-up, snow/ice, barren/sparsely vegetated, and water bodies. We also filter out potential fire-affected data by identifying

for each day the cells of the $0.25° \times 0.3125°$grid in which fires are reported in the MODIS active fire product. The data from these, and the adjacent cells, are then removed for that day as well as the preceding and succeeding days, following (Barkley et al., 2013).

Second, we determine the model relationship between local isoprene emissions $E$ (molec/cm$^2$/s), as calculated by MEGAN (Guenther et al., 2012), and HCHO columns $\Omega$ (molec/cm$^2$):

$$\Omega = S E_{\mathrm{VOC}} + \Omega_b,\tag{2}$$





where the slope $S$ represents the production of HCHO column per emission of isoprene, and the intercept $\Omega_b$ represent the HCHO column contributions from longer-lived VOCs mainly from the oxidation of methane. To estimate isoprene emissions that are consistent with the HCHO columns we transpose this linear relationship.

For both study regions, we find a statistically significant linear relationship between those variables (Table 3), where Pearson correlation coefficients $r$ range from 0.52 to 0.82, with a typical value in excess of 0.70. The slope values ($10^3$s) vary with region and season. The offsets that represent the background HCHO column is higher during the pre-monsoon and monsoon summer months when we expect larger HCHO production from a range of longer-lived VOC (including $CH_4$) due to higher
values of OH.

   Our *a posteriori* emission estimates are generally lower than *a priori* values, reflecting the positive model HCHO column bias. This is most pronounced over the Northeast during the monsoon season, where *a posteriori* isoprene emissions are 88% lower than *a priori* estimate, due to the model not capturing the sharp observed decline in HCHO that appears to be linked with monsoon conditions.

**4   Concluding Remarks**

We used models of atmospheric chemistry to interpret HCHO column distributions during 2014 observed by the Ozone Monitoring Instrument (OMI) satellite instrument over India. The annual mean OMI distribution of clear-sky HCHO columns is dominated by a distinctive meridional gradient in the northern half of the country, and by localized regions of high columns that coincide with
forests. We found that the nested GEOS-Chem atmospheric chemistry model (spatially resolved at $\simeq$25 km) reproduces these broad-scale observed features with a positive model bias, particularly over the Indo-Gangetic Plain and Delhi.

   Over India, HCHO has biogenic, pyrogenic, and anthropogenic sources of volatile organic compounds (VOCs), some of which are spatially and temporally disaggregated. Using the CAABA 0-D
photochemistry model, we explored a range of forest and urban photochemical environments found over India and their subsequent influence on HCHO concentrations. HCHO columns are related to local VOC emissions with a spatial smearing that increase with the VOC lifetime. We found that isoprene has the largest molar yield of HCHO that is typically realized within a few hours in the presence of moderate levels of nitrogen oxides ($\simeq$1 ppbv), in agreement with previous studies. How-
ever, we also found that forested regions that neighbour major urban conurbations (e.g. in the state of Kerala) are exposed to much higher levels of nitrogen oxides ($\simeq$8 ppbv). This results in depleted hydroxyl radical concentrations and a delay in the production of HCHO from isoprene oxidation. Informed by a regional bottom-up emission inventory for India, we found that propene is the only major component of anthropogenic VOCs that produces HCHO at comparable (but slower) rate to
isoprene.


We found that the GEOS-Chem model reproduces observed spatial distributions during winter (JF) and pre-monsoon months (MAM) better than during monsoon (JJAS) and post-monsoon (OND) months. We attributed these differences in model skill to the response of the natural biosphere to changes in the meteorological and photochemical environments associated with the onset and retreat

of the monsoon. We found that on a continental scale much of the seasonal cycle in observed HCHO columns can be explained by monthly variations in surface temperature. This observation together with the strong local relationship we found between isoprene nemissions and HCHO production suggests a role for biogenic VOC, in agreement with the GEOS-Chem model calculation. We also found that the seasonal cycle during 2014 is not significantly different from the 2008-2015 mean

seasonal variation but there are large year to year variations. There are two main loci for biomass burning (states of Punjab and Haryana, and northeastern India), which we found only contributes a significant contribution to observed columns during March to April over northeastern India. The slow production of HCHO from propene oxidation results in a smeared hotspot over Delhi that we resolved only by using a temporal oversampling method. Based on comparing GOME-2 and OMI

HCHO column distributions, we proposed an argument that the early morning overpass time is better for quantifying anthropogenic emissions soon after the rush hour and before biogenic emissions are at their early afternoon peak.

    Using a linear regression model to relate GEOS-Chem isoprene emissions to HCHO columns we inferred seasonal isoprene emissions over two key forest regions from the OMI HCHO column

data. We found that the *a posteriori* emissions are typically lower than the *a priori* emissions, with a much stronger reduction of emissions during the monsoon season. This reduction in emissions during monsoon months coincided with a large drop in satellite observations of leaf phenology. Large-scale differences in observed and model HCHO columns during monsoon months may highlight errors in seasonal variations in basal emission rates and/or model errors associated with the

underlying meteorological environments, e.g. partitioning of direct and diffuse photosynthetically active radiation.

    The next logical step to this analysis is to estimate simultaneous estimates of anthropogenic, pyrogenic, and biogenic VOC emissions by using data collecting data from morning and afternoon local overpass times. In the case of biogenic VOC emissions, information from HCHO columns together

with leaf phenology (e.g. leaf area index) and land surface parameters (e.g., soil moisture), can be integrated to develop a new satellite data-driven isoprene emission inventory. A self-consistent pan-tropical emission inventory for isoprene, for example, would help to improve understanding of tropospheric $O_3$ and organic aerosol that represent some of the largest uncertainties associated with the Earth system. Achieving this capability is greatly enhanced by the launch of TROPOMI that will

result in daily maps of HCHO columns and complementary trace gases at a spatial resolution of 7 km, which dramatically increases the number of clear-sky scenes available for the analysis.





*Acknowledgements.* L.S. was funded by the NERC National Centre for Earth Observation (NCEO020005), and P.I.P. gratefully acknowledges his Royal Society Wolfson Research Merit Award. L.S. also acknowledges funding from the British Council Newton Fund (215829867), administered by the University of Birmingham.

We are grateful to Kelly Chance from the Harvard-Smithsonian Center for Astrophysics, the Harvard University GEOS-Chem group who maintains the model, and to Rolf Sander for maintaining the CAABA/MECCA box model.

*Author contributions.* L.S. and P.I.P designed the computational experiments; P.I.P. and L.S. wrote the paper; G.G.A. provided input on the paper regarding the OMI data analysis.





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



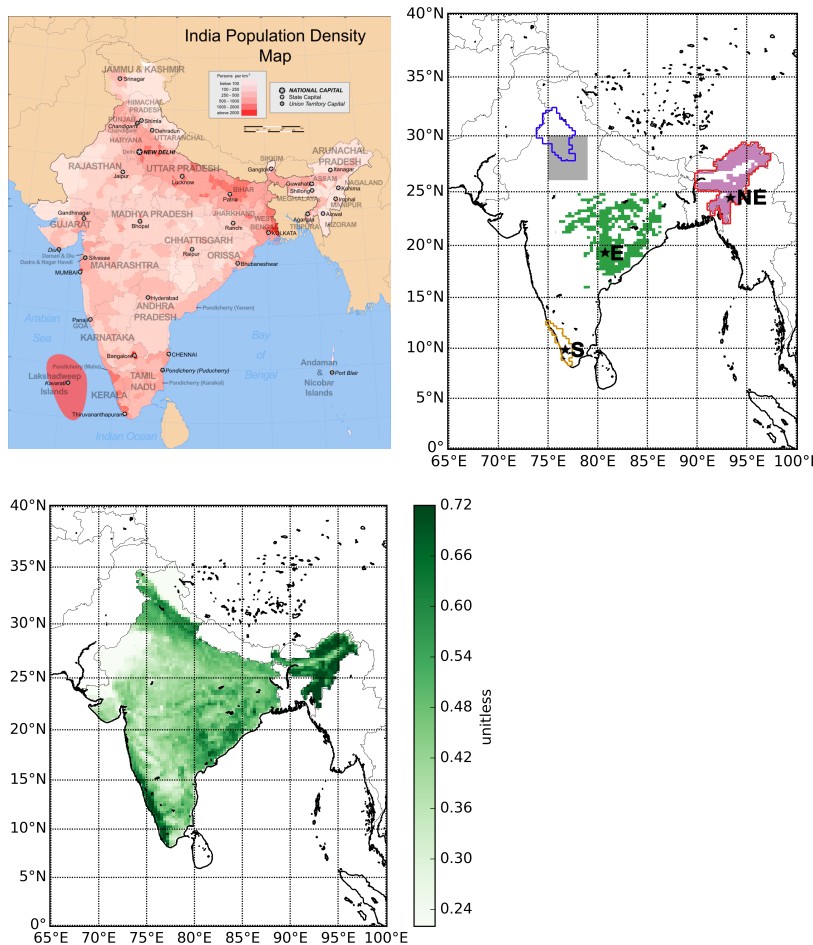

Figure 1: Geography of India: population density based on the 2001 census (top left); focal points of this study (top right); and annual mean values of the Normalized Difference Vegetation Index (NDVI) for 2014 from the NASA MODIS instrument, averaged on the $0.25° \times 0.3125°$ GEOS-Chem nested grid. For the map of focal points: the blue outline represent the states of Punjab and Haryana; the red outline denotes Seven sister states; and the yellow outline represents the state of Kerala. The pink and green shaded areas denote the NE and E forest sites, where infer isoprene emissions from OMI HCHO columns. The grey shaded area denote the oversampling region used to study Delhi. The stars denote the sites where we study HCHO production using a 0-D photochemical model. The population image is taken from Wikimedia c/o CC-by-sa PlaneMad/Wikimedia:https://commons.wikimedia.org/wiki/File:India_population_density_map_en.svg.





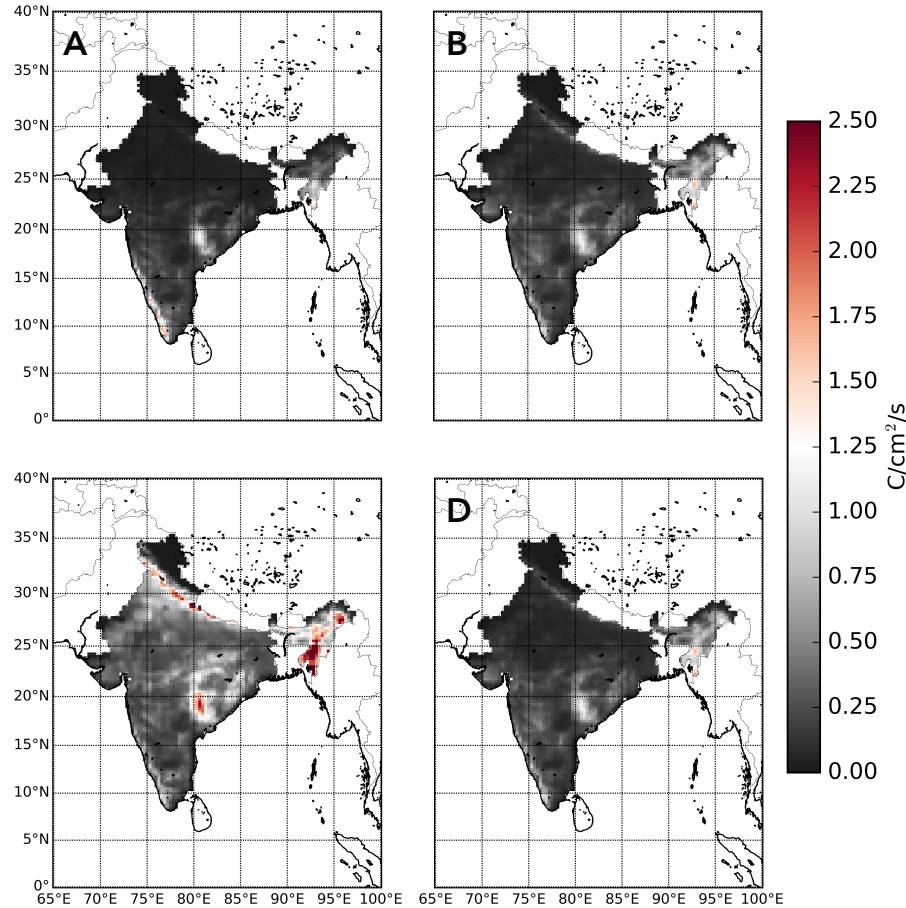

Figure 2: Seasonal isoprene emission rates (C/cm$^2$/s) from the MEGAN inventory (Guenther et al., 2012). Data are described on the GEOS-Chem $0.25° \times 0.3125°$ grid for A) winter (JF), B) pre-monsoon (MAM), C) monsoon (JJAS), and D) post-monsoon seasons (OND).





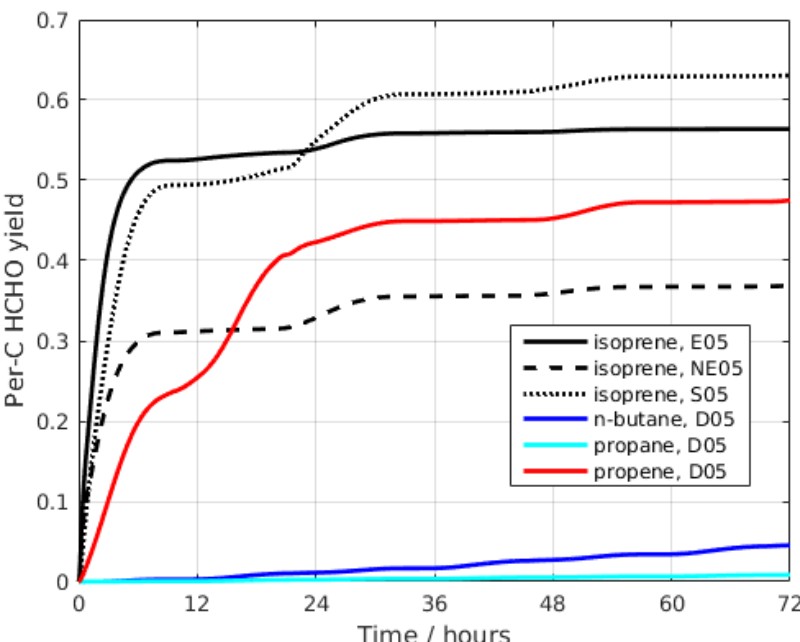

Figure 3: Time-dependent cumulative HCHO yield (per unit C) produced from isoprene in three contrasting photochemical environments, and from propene, propane, and n-butane in an urban photochemical environments based on Delhi. Calculations are denoted by an alphanumeric code that is explained in Table 1.



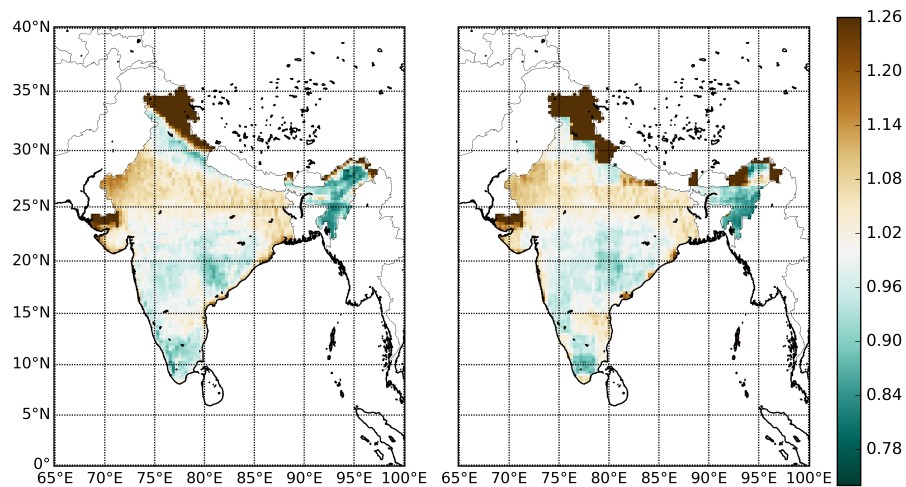

Figure 4: Annual mean air mass factors (AMFs) for 2014 calculated using (left) nested $0.25° \times 0.3125°$ and (right) standard $2.0° \times 2.5°$ GEOS-Chem grid.



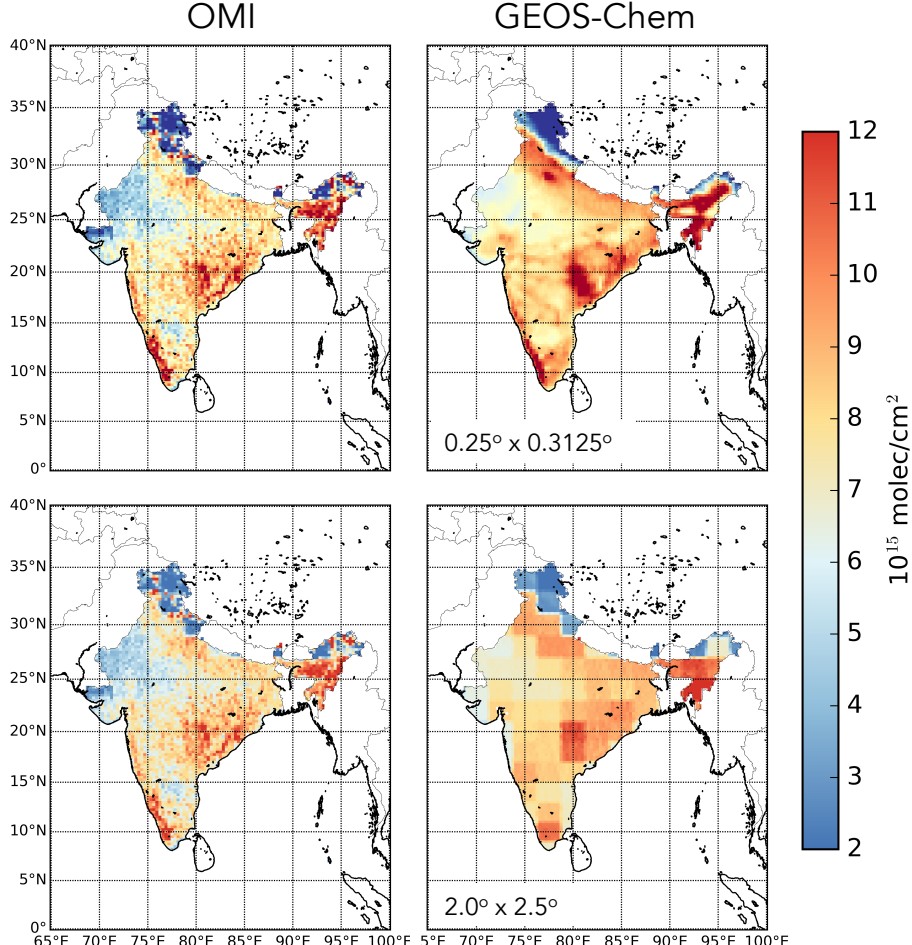

Figure 5: Annual mean clear-sky (left) OMI and (right) GEOS-Chem HCHO vertical columns ($10^{15}$ molec/cm$^2$) over India for 2014. The OMI vertical columns are described on the $0.25° \times 0.3125°$ GEOS-Chem grid. These vertical columns are transformed from observed slant columns using AMFs informed by the distribution of HCHO described by the (top) nested $0.25° \times 0.3125°$ and (bottom) standard $2.0° \times 2.5°$ GEOS-Chem grid. Data exclusion criteria are described in the main text. The model is sampled at the time and location of each observed scene.



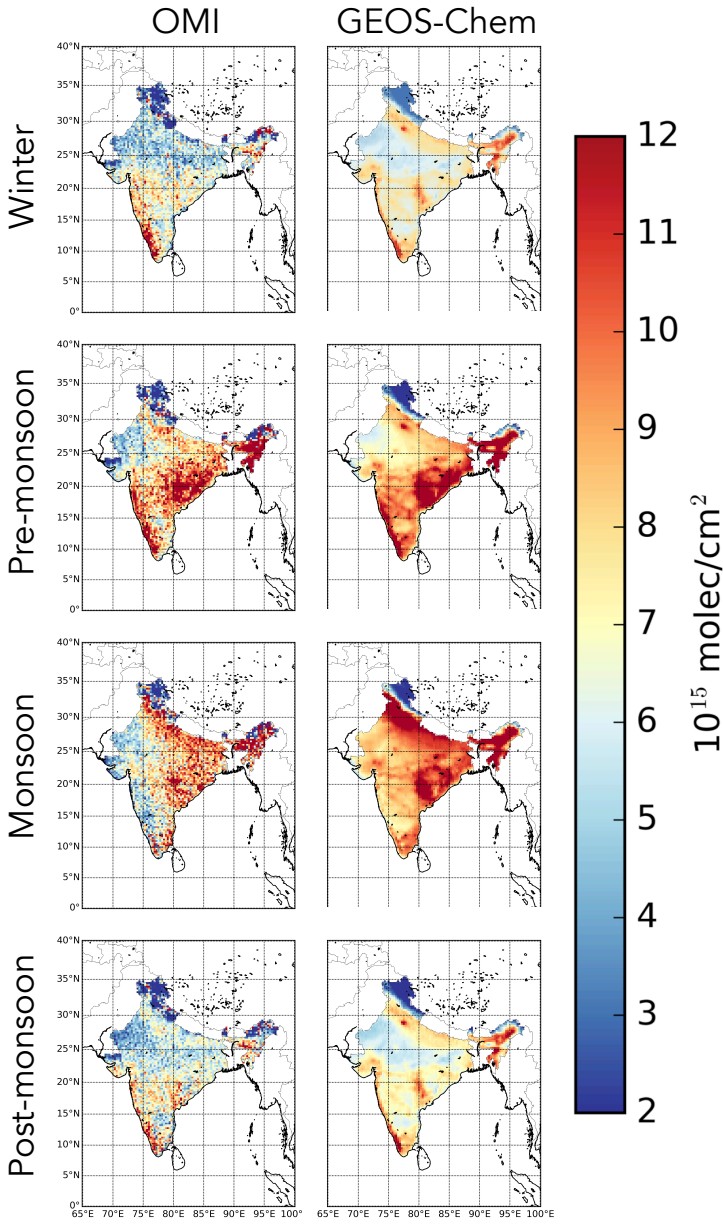

Figure 6: Seasonal mean clear-sky (left) OMI and (right) GEOS-Chem HCHO vertical columns ($10^{15}$ molec/cm$^2$) for 2014, averaged on a common $0.25° \times 0.3125°$ grid. Seasonal definitions are described in Section 2.





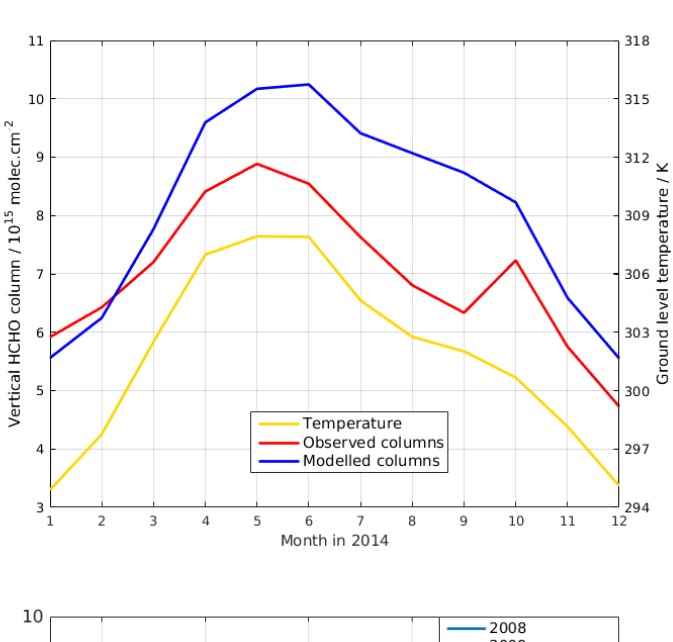

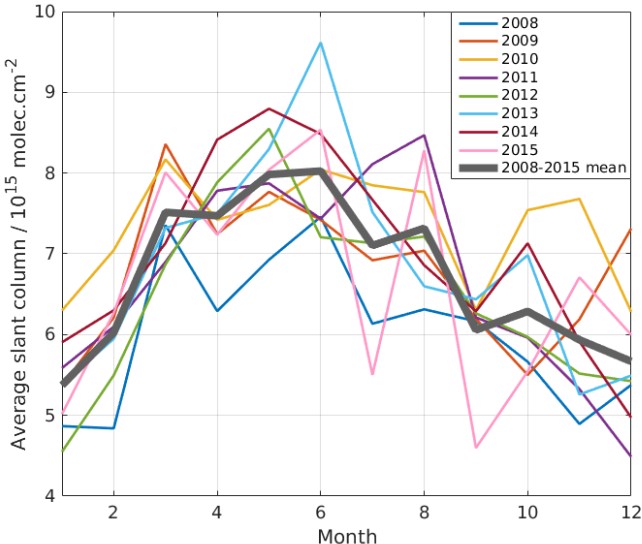

Figure 7: Timeseries of (top) OMI and GEOS-Chem vertical columns ($10^{15}$ molec/cm$^2$) for 2014, and (bottom) OMI slant columns ($10^{15}$ molec/cm$^2$) from 2008 to 2015 over India. GEOS-FP ground-level temperature (K) is also shown in the left panel. The thick grey line in the right panel denotes 2008–2015 HCHO column monthly means.



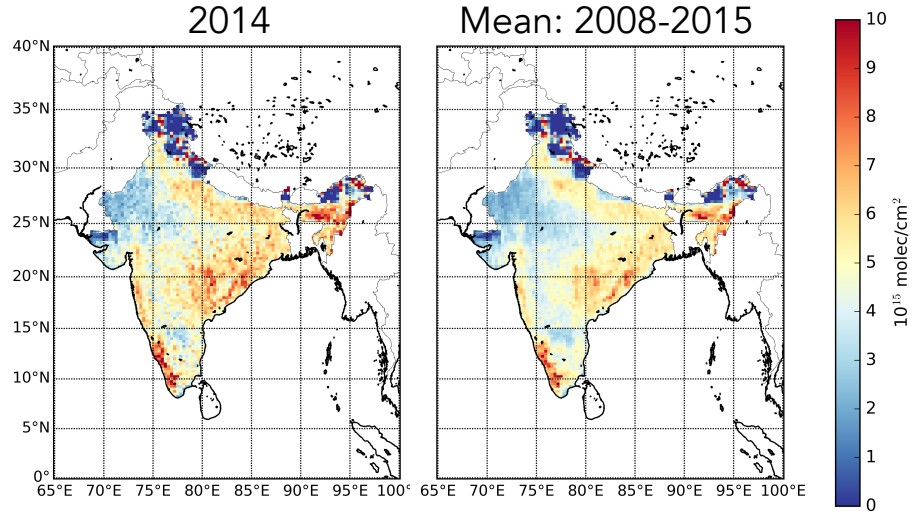

Figure 8: Annual mean OMI HCHO slant columns ($10^{15}$ molec/cm$^2$) over India for (left) 2014 and (right) 2008–2015, averaged on a common $0.25° \times 0.3125°$ grid.





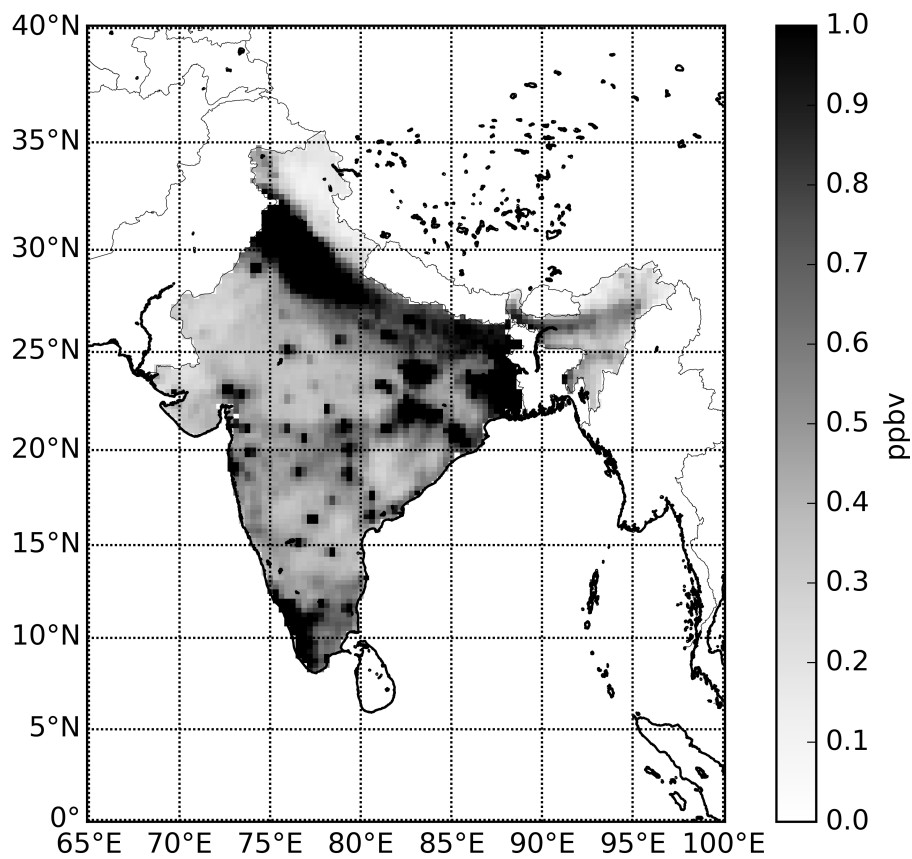

Figure 9: Annual mean GEOS-Chem ground-level NO$_x$ mixing ratios (ppbv) in 2014, averaged on the $0.25° \times 0.3125°$ grid.





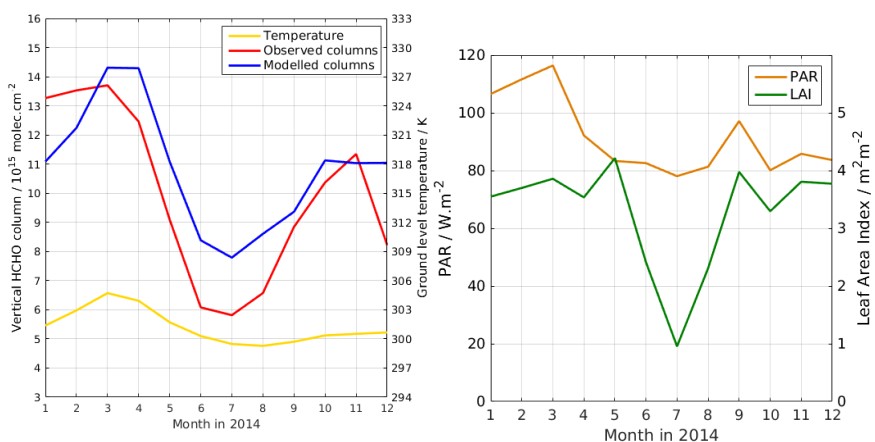

Figure 10: Timeseries of (left) monthly mean OMI and GEOS-Chem vertical HCHO columns ($10^{15}$ molec/cm$^2$), and GEOS-FP ground-level temperature, and (right) PAR (W/m$^2$) and LAI (m$^2$/m$^2$) over the state of Kerala in 2014.



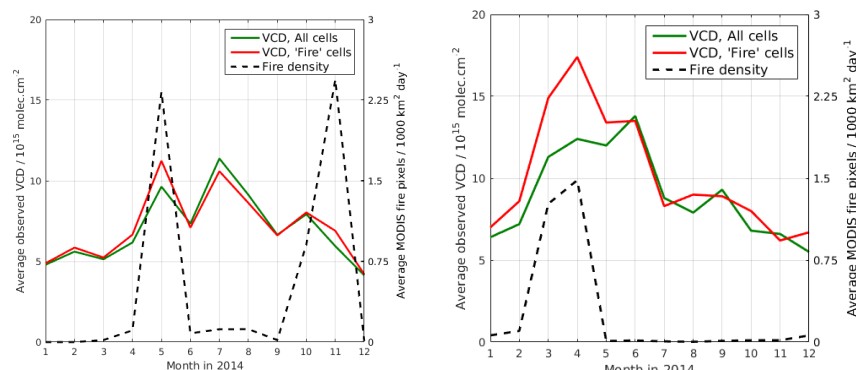

Figure 11: Timeseries of monthly mean HCHO columns ($10^{15}$ molec/cm$^2$) and the number of MODIS fire detections ($10^3$ km$^2$/day) over (left) the states of Punjab and Haryana and (right) North-eastern India. The green line denotes all HCHO columns over the region, and the red line denotes columns that are most affected by fires as indicated by MODIS firecounts.




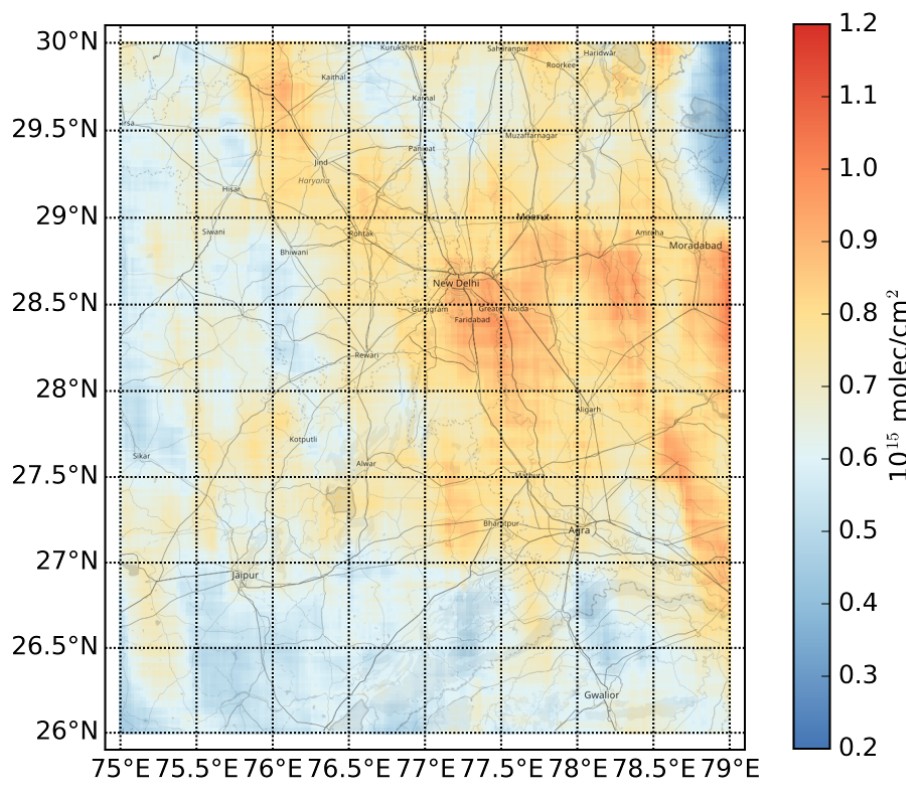

Figure 12: Oversampled distribution of OMI vertical HCHO columns ($10^{15}$ molec/cm$^2$) around Delhi. Underlying map imagery is taken from openstreetmap.org





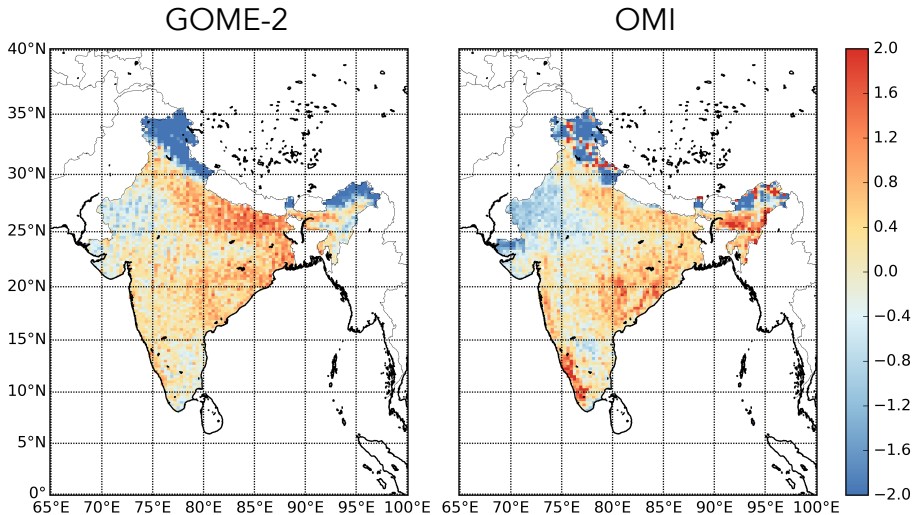

Figure 13: Standardized HCHO slant columns for 2014 (unitless) from (left) the Global Ozone Monitoring Experiment-2 (GOME-2) aboard MetOp and (right) OMI, averaged on a common $0.25° \times 0.3125°$ grid. GOME-2 and OMI have local equatorial overpass times of 0930 and 1330, respectively.





Table 1: Photochemical and meteorological scenarios used for box model calculations. The embold-
ened letters E, NE, S denote the forested regions (Figure 1) and D denotes Delhi. Ground-level
temperature, relative humidity, and boundary layer height are denoted by T, RH, and BLH.

| Scenario name | Location | Month of year | T [K] | RH [%] | BLH [m] | Fixed mixing ratio [ppbv] | | |
|---|---|---|---|---|---|---|---|---|
| | | | | | | O$_3$ | CO | NO$_2$ |
| E01 | **E** 19.3°N 80.6°E | Jan | 295 | 56 | 1240 | 49 | 232 | 1.1 |
| E05 | | May | 307 | 33 | 1873 | 63 | 131 | 2.4 |
| E08 | | Aug | 299 | 84 | 685 | 39 | 114 | 1.3 |
| E11 | | Nov | 304 | 64 | 611 | 49 | 220 | 0.8 |
| NE01 | **NE** 24.4°N 93.1°E | Jan | 292 | 65 | 648 | 42 | 277 | 1.8 |
| NE05 | | May | 302 | 59 | 1202 | 55 | 154 | 0.8 |
| NE08 | | Aug | 299 | 88 | 666 | 24 | 127 | 1.3 |
| NE11 | | Nov | 296 | 73 | 925 | 43 | 241 | 1.4 |
| S01 | **S** 9.8°N 76.6°E | Jan | 300 | 59 | 1090 | 48 | 260 | 8.9 |
| S05 | | May | 301 | 78 | 725 | 32 | 146 | 8.6 |
| S08 | | Aug | 299 | 84 | 690 | 18 | 122 | 8.1 |
| S11 | | Nov | 299 | 77 | 732 | 45 | 274 | 8.9 |
| D05 | **D** 28.61°N 77.23°E | May | 307 | 18 | 1975 | 37 | 2300 | 19 |





Table 2: Results of the photochemical box modelling for forested region scenarios (Table 1).

| Scenario | Isoprene lifetime | Time to reach peak HCHO signal | HCHO yield |
|---|---|---|---|
| | [min] | [min] | [per unit C] |
| E01 | 26 | 79 | 0.50 |
| E05 | 14 | 59 | 0.55 |
| E08 | 10 | 33 | 0.49 |
| E11 | 20 | 83 | 0.41 |
| NE01 | 28 | 89 | 0.59 |
| NE05 | 10 | 33 | 0.38 |
| NE08 | 10 | 33 | 0.58 |
| NE11 | 18 | 57 | 0.53 |
| S01 | 64 | 127 | 0.63 |
| S05 | 48 | 147 | 0.63 |
| S08 | 70 | 189 | 0.66 |
| S11 | 52 | 153 | 0.62 |




Table 3: *A priori* and *a posteriori* isoprene emission estimates ($10^{11}$C/cm$^2$/s) over NE and E forest sites (Figure 1), and the model linear regression coefficients that relate model isoprene emissions and HCHO columns.

| East Forest Region | | | | | |
|---|---|---|---|---|---|
| Season | Mean *a priori* emission [$10^{11}$C/cm$^2$/s] | Slope $S$ [$10^3$ s] | Intercept $\Omega_b$ [$10^{15}$ molec/cm$^2$] | $r^2$ | Mean a posteriori emission [$10^{11}$C/cm$^2$/s] |
| Winter | 2.8 | 4.1 | 5.3 | 0.59 | 3.0 |
| Pre-monsoon | 10.2 | 4.3 | 7.0 | 0.67 | 7.8 |
| Monsoon | 7.4 | 4.3 | 7.6 | 0.51 | 3.3 |
| Post-monsoon | 3.6 | 4.9 | 5.2 | 0.47 | 2.6 |
| Northeast Forest Region | | | | | |
| Winter | 3.5 | 3.9 | 3.9 | 0.42 | 4.4 |
| Pre-monsoon | 17.9 | 2.9 | 6.2 | 0.58 | 12.6 |
| Monsoon | 14.6 | 2.2 | 8.7 | 0.27 | 1.7 |
| Post-monsoon | 6.6 | 4.9 | 5.1 | 0.49 | 2.7 |