# Peer review of "Which processes drive observed variations of HCHO columns over India?"

_Atmospheric Chemistry and Physics, 2017_

## Referee Comment (RC1) · Anonymous Referee #1 · 29 Nov 2017

**Surl et al., 2017, ACP, Which processes drive observed variations of HCHO columns over India?**

**General Description of manuscript:**

The authors use total column formaldehyde measurements from the Ozone Monitoring Instrument to assess spatial and temporal variability of volatile organic compounds over India. They identify that isoprene oxidation leading to the formation of formaldehyde is the largest local source contributor to formaldehyde over India and that anthropogenic emissions of non-volatile organic compounds can only be spatially resolved over Delhi with oversampling. The content of this manuscript is appropriate for ACP. The authors have taken careful consideration of the unique seasonality and spatial features over India, but there are quite a few editorial errors that can be eliminated with a careful read-through. Some of these are identified below, along with general comments and specific minor changes.

**General Comments:**

Indicate the locations of geographic features in India on the Figures provided so that it is easy for someone not familiar with the country to follow along. For example, locations of the Thar Desert, the IGP, and the five megacities should be provided in Figure 1 (Lines 50-52), the features referred to in lines 310-315 should be indicated in Figure 5, and the features referred to in lines 292-293 should be located on the plot (in particular Rann of Kutch).

In the list of methods applied to deriving isoprene emissions (Lines 85-90), the local $NO_x$-dependent relationship used in Marais et al. (2012) in missing. The authors point out that the correlations and slopes obtained in each season between HCHO and isoprene emissions in GEOS-Chem are highly variable. What drives this variability? Is it $NO_x$? Would it be more effective then to use a $NO_x$-dependent HCHO yield to derive isoprene emissions from OMI HCHO?

Why when comparing the multiyear average and year-to-year variability of OMI HCHO to the 2014 values is OMI HCHO for 2008-2015 used? Why not use the full OMI HCHO record starting from the first complete year (i.e. 2005-2015)?

What drives the year-to-year variability in HCHO (Figure 7, Line 22)?

Some information regarding the isoprene oxidation chemistry in each model is needed so that the reader can assess whether the 2 models (CAABA and GEOS-Chem) have the same isoprene oxidation chemistry, or whether CAABA, due to lower computational needs than a 3D model, has a more detailed representation of isoprene oxidation than GEOS-Chem. Have both models kept pace with new findings of isoprene oxidation mechanisms reported in the literature? Are $NO_x$-dependent HCHO yields from isoprene oxidation similar for the 2 models?

The manuscript is missing a Data Availability section (https://www.atmospheric-chemistry-and-physics.net/about/data_policy.html).

**Specific Comments:**

Line 11: "comparable (slower)" is contradictory. Which is it?

Lines 69-71: Please specify whether these trends in HCHO are for large urban areas, rural regions or across the country.

Line 73: "By" is missing in "supported … detailed modelling studies".

Line 86: Close bracket missing.

Line 139: "detailed" should be "detail".

Lines 163-164: Incorrect in-text referencing style for Sander et al., 2014.

Line 186: Should "signal" be "sigma"?

Line 186-187: "30 in the troposphere" is appropriate for a fixed tropopause, but doesn't the model have a dynamic tropopause that leads to variability in the number of levels in the troposphere?

Line 194: Remove redundant "of".

Line 195: Is MIX a mosaic of regional inventories? As written this isn't clear.

Line 201: Provide the version of MEGAN that is used.

Lines 206-207: Say where the drivers of MEGAN are from. Is LAI from the model fields or from MODIS?

Line 297: The meaning of the 11% is not clear. Is 11% the percent increase in vertical columns relative to slant columns after applying the AMF? Or is this an increase in the spatial correlation after applying the AMF?

Lines 347-348: Reiterate that the slant columns are used for 2008-2015.

Line 351: "that" should be removed in "shows that".

Line 352: Should "Median" be mean? That's what's shown in Figure 8.

Line 384: Don't mature leaves emit more isoprene than young and senescing leaves (Guenther et al., 2006)? This sentence needs to be revised to reflect that.

Line 404: "thes" is a typo.

Lines 424-425: Why does the bottom-up emission inventory overestimate emissions from the transport sector when the inventory is for 2010 and the simulation year is 2014? Wouldn't this misrepresent the growth in this sector and so underestimate emissions from the transport sector? The reasoning for an overestimate is not immediately clear.

Line 456: Are the "sample mean" and "sample standard deviation" the regional mean and standard deviation of column HCHO for the whole country? If so, please clarify this in the text.

Line 461: Rather than stating that further work is "outside the scope of this current study", more impactful to point out that this is an early demonstration of the efficacy of future geostationary satellites that will provide constraints on the temporal variability of HCHO.

Line 477: Incorrect in-text citation style for Barkley et al., 2013.

Figure 1:
- City names (perhaps capital cities?) and the legend in the population map are not legible.
- The population map has a panel header ("India Population Map"). The other panels would benefit from headers too.
- The red outline could be made clearer in the top right map.

Figure 2:
- Label "C" is missing.
- Consider choosing a colour bar with a more dynamic colour range. As is, JJAS emissions seem only marginally higher than emissions in other months.
- What are the isoprene emission units? "g C", "molecules C", or "mg C"?

Figure 3:
- Consider including initial $NO_x$ (or NO) concentrations for each isoprene HCHO yield line shown in the plot.

Figure 7:
- "Left" should be "top" and "right" should be "bottom" in the figure caption.

Figure 9:
- Consider choosing a colour bar with a more dynamic colour range.

Figure 12:
- Labels on the map are not legible.
- Consider replacing "map imagery" with "road network".

Table 1:
- Include a footnote in the table to indicate the source of variables (T, RH, BLH, $O_3$, CO, $NO_2$) provided.

**References:**

Guenther et al., 2006, Estimates of global terrestrial isoprene emissions using MEGAN (Model of Emissions of Gases and Aerosols from Nature), Atmos. Chem. Phys., 6, 3181–3210, 2006.

Marais et al., 2012, Isoprene emissions in Africa inferred from OMI observations of formaldehyde columns, Atmos. Chem. Phys., 12, 6219-6235, 2012.

---

## Referee Comment (RC2) · Anonymous Referee #2 · 13 Dec 2017

This manuscript provides an in-depth and comprehensive analysis of OMI HCHO columns over India. To answer the question posed by the manuscript title, HCHO columns over India are driven largely by biogenic isoprene, with smaller enhancements from anthropogenic VOCs and pyrogenic activity. The authors use a box model analysis to analyze the potential contribution of various VOCs to observed HCHO columns. The spatial and temporal patterns of HCHO columns were compared against results from GEOS-Chem. Isoprene emissions derived from the linear approach suggest the effects of monsoons may not be well captured.

Below, I suggest a few adjustments which would improve the analysis, mostly relating to the isoprene-HCHO relationship as discussed in the manuscript. In general, the manuscript is well written and will be of high interest to readers, and I recommend

publication after minor revisions.

Major comments:

1.) Verison 10-01 of the GEOS-Chem model is used in this analysis. The mechanism in v11-01 (released Feb 2017) includes large revisions on the isoprene oxidation mechanism effecting the HCHO yield. Specifically, the prompt yield of HCHO from the reaction of the isoprene peroxy radical with NO (RIO2 + NO) is increased by 24% (from 0.66 to 0.82). The revised mechanism is in good agreement with fully explicit mechanisms and generally reproduces observations (Marvin et al., 2017). Because we have more confidence in the updated mechanism, and because the mechanism is likely to affect the modeled HCHO columns shown here, the updated isoprene oxidation mechanism should be used in this analysis.

2.) It is unclear here if the chemical mechanism used in the CABBA/MECA box model is the same as used in the GEOS-Chem simulation. Ideally, the same mechanisms would be used.

3.) Recently, Zhu et. al (2016) found that the OMI SAO HCHO product used here was biased low by 37% compared to observations over the Southeastern United States. Do we have any reason to suspect that this bias may persist in other regions of the globe? I understand that in-situ HCHO observations are likely limited for this region; however, acknowledgement of the potential for bias and the need for validation is warranted.

Minor comments:

Page 3, line 79: It is unclear what is meant by "this approach".

Page 3, sentence starting at line 88: Wolfe et al. (2016) is cited as a reference for the uncertainty in HCHO production in the low NOx regime. In contrast, I think this paper suggests that the NOx dependency of HCHO production is well captured by updated chemical mechanisms. In any case, it is not clear to me why uncertainty in VOC emissions and in the low NOx yield means "both approaches provide useful

insights". Perhaps it is more useful to point out that while the inverse model approach gives a more rigorous result, it comes at a higher computational expensive, and the local linear approach provides a useful approximation where there is little smearing.

Page 7, line 201: List which version of MEGAN.

Page 7, line 213: GEOS-Chem HCHO columns are averaged between 1300 and 1500 to provide comparison with OMI observations. Is this a hold-over from GEOS-Chem simulations run at coarse resolution with longer (60 min) timesteps? At the 10 minute chemistry timestep often used for $0.25° \times 0.3125°$ simulations, I would have expected the output to be averaged over a smaller time frame (1300 to 1400). Perhaps the effect is small, but Is there a reason for the $1300 - 1500$ averaging window? What are the chemistry and transport time steps used in this simulation?

Page 8, lines 240-226. The box model analysis of isoprene oxidation as a function of NOx seems redundant with other recent publications, which also have the benefit of comparison to in-situ observations (Wolfe et al., 2016; Marvin et al., 2017). For example, figure 3 is reminiscent of Figure S3 in Marvin et al. 2017. At the very least, the box model results here should be compared with those in previous literature.

Page 11, line 370: add a section refence (i.e. "as described in section 2.2")

Page 15, lines 491: The procedure of deriving a posteriori emissions using the linear approach should be briefly described for readers unfamiliar with the process.

Page 38, Table 3: Consider adding a "% change" column

Typographical errors:

In general, several subsections are titled but lack numbers (e.g., "Box modeling" in section 2.1). Sub-sub sections may be useful to the reader.

Page 8, line 253: replace "isoprene getting oxidized" with "isoprene oxidation"

Page 13, line 429: should read "collected"

Page 14 line 471: should read "16-25°N and"

Page 14 line 472: should read "°E ("". Also, replace "We remove scenes that correspond to" with "We remove scenes with"

Page 16 line 522: should read "emissions"

Page 23, Figure 1 caption, line 6: should read "where we infer"

Page 23, Figure 1 caption, line 7: should read "area denotes"

References:11 Marvin, M. R., Wolfe, G. M., Salawitch, R. J., Canty, T. P., Roberts, S. J., Travis, K. R., Aikin, K. C., de Gouw, J. A., Graus, M., Hanisco, T. F., Holloway, J. S., Hübler, G., Kaiser, J., Keutsch, F. N., Peischl, J., Pollack, I. B., Roberts, J. M., Ryerson, T. B., Veres, P. R., and Warneke, C.: Impact of evolving isoprene mechanisms on simulated formaldehyde: An inter-comparison supported by in situ observations from SENEX, Atmos. Environ., 164, 325-336, http://dx.doi.org/10.1016/j.atmosenv.2017.05.049, 2017.

Wolfe, G. M., Kaiser, J., Hanisco, T. F., Keutsch, F. N., de Gouw, J. A., Gilman, J. B., Graus, M., Hatch, C. D., Holloway, J., Horowitz, L. W., Lee, B. H., Lerner, B. M., Lopez-Hilifiker, F., Mao, J., Marvin, M. R., Peischl, J., Pollack, I. B., Roberts, J. M., Ryerson, T. B., Thornton, J. A., Veres, P. R., and Warneke, C.: Formaldehyde production from isoprene oxidation across NOx regimes, Atmos. Chem. Phys., 16, 2597-2610, doi:10.5194/acp-16-2597-2016, 2016.

Zhu, L., Jacob, D. J., Kim, P. S., Fisher, J. A., Yu, K., Travis, K. R., Mickley, L. J., Yantosca, R. M., Sulprizio, M. P., De Smedt, I., Abad, G. G., Chance, K., Li, C., Ferrare, R., Fried, A., Hair, J. W., Hanisco, T. F., Richter, D., Scarino, A. J., Walega, J., Weibring, P., and Wolfe, G. M.: Observing atmospheric formaldehyde (HCHO) from space: validation and intercomparison of six retrievals from four satellites (OMI, GOME2A, GOME2B, OMPS) with SEAC(4)RS aircraft observations over the southeast US, Atmos. Chem. Phys., 16, 13477-13490, doi:10.5194/acp-16-13477-2016, 2016.

---

## Author Response (AR1)

**Author Response to Reviews of**

**Which processes drive observed variations of HCHO columns over India?**

Luke Surl, Paul I. Palmer, and Gonzalo González Abad
*Atmospheric Chemistry and Physics Discussions,* `doi:10.5194/acp-2017-1017`
* * *
RC: *Reviewer Comment*,    AR: *Author Response*,    □ Manuscript text

We thanks the reviewers for their careful readings of the manuscript and suggestions for improvement. This document details our responses to these and the subsequent changes we have made to the revised manuscript.

**1. Reviewer #1**

**1.1. General comment #1**

RC: *Indicate the locations of geographic features in India on the Figures provided so that it is easy for someone not familiar with the country to follow along. For example, locations of the Thar Desert, the IGP, and the five megacities should be provided in Figure 1 (Lines 50-52), the features referred to in lines 310-315 should be indicated in Figure 5, and the features referred to in lines 292-293 should be located on the plot (in particular Rann of Kutch).*

AR: *We have added labels to these figures to assist the reader in locating these features.*

**1.2. General comment #2**

RC: *In the list of methods applied to deriving isoprene emissions (Lines 85-90), the local NOx dependent relationship used in Marais et al. (2012) in missing. The authors point out that the correlations and slopes obtained in each season between HCHO and isoprene emissions in GEOS-Chem are highly variable. What drives this variability? Is it NOx? Would it be more effective then to use a NOx-dependent HCHO yield to derive isoprene emissions from OMI HCHO?*

AR: *Not including Marais et al (2012) here was an egregious oversight. At least some of the uncertainty in our linearized isoprene emissions will have been caused by variations in HCHO yields due to its sensitivity to NOx. Certainly, over India our understanding of NOx emissions is highly uncertain so we felt it was not appropriate to use this method. Nevertheless, we have included text to acknowledge and explain the benefits of the Marais method:*

> *For central Africa, Marais et al (2012) employed an inversion method that accounted for the NOx-dependence of the isoprene-HCHO relationship*

**1.3. General comment #3**

RC: *Why when comparing the multiyear average and year-to-year variability of OMI HCHO to the 2014 values is OMI HCHO for 2008-2015 used? Why not use the full OMI HCHO record starting from the first complete year (i.e. 2005-2015)?*

AR: *The purpose of this exercise was not to show the whole time series but rather to show that 2014 is representative of the previous few years. As Mahahjan et al 2015 showed that there was a long-term trend to the HCHO columns over India, we determined extending this comparison further back had the potential for causing confusion.*

**1.4. General comment #4**

RC: ***What drives the year-to-year variability in HCHO (Figure 7, Line 22)?***

AR: *This is a good question but not the main focus of this current analysis. We wanted to understand whether our study year (2014) was representation of a longer time period. It seems from our analysis that 2014 was not anomalous. However, based on our analysis it seems that surface temperature is a good predictor for continental-scale variations of OMI slant columns of HCHO. For a more detailed analysis of the drivers we refer the reader to Mahahjan et al 2015.*

> *Mahajan et al. (2015) reports an analysis of the drivers of year-to-year variations in HCHO columns over India.*

**1.5. General comment #5**

RC: ***Some information regarding the isoprene oxidation chemistry in each model is needed so that the reader can assess whether the 2 models (CAABA and GEOS-Chem) have the same isoprene oxidation chemistry, or whether CAABA, due to lower computational needs than a 3D model, has a more detailed representation of isoprene oxidation than GEOS-Chem. Have both models kept pace with new findings of isoprene oxidation mechanisms reported in the literature? Are NOx-dependent HCHO yields from isoprene oxidation similar for the 2 models?***

AR: *The mechanisms are not the same. We have added text referring the reader to studies where both GEOS-Chem and MIM2 (the mechanism used in the box model) are compared to the Master Chemical Mecahnism and their HCHO-related chemistry results are assessed.*

> The chemical mechanisms from the CAABA box model and the GEOS-Chem model are not identical. CAABA includes a more explicit chemical mechanism and a reduced version of the Mainz Isoprene Mechanism 2 (MIM2, Taraborelli et al, 2009). The version of GEOS-Chem we use here is described by Eastham et al, 2014. The latest version of the GEOS-Chem chemical mechanism, released after the majority of our calculations had been completed, results in an increase of the HCHO yield from the oxidation of isoprene. We find using preliminary calculations with v11.01 that the revised mechanism does not systematically change the results shown here. Previous studies have evaluated the performance of GEOS-Chem (Marvin et al, 2017) and MIM2 (Taraborelli et al, 2009) against the Master Chemical Mechanism. The GEOS-Chem v10.01 mechanism slightly underestimates HCHO production in high-NOx conditions and the MIM2 mechanism shows similar HCHO production across a wide range of NOx values.

**1.6. General comment #6**

RC: ***The manuscript is missing a Data Availability section (`https://www.atmospheric-chemistry-and-physics.net/about/data_policy.html`).***

AR:    *We have added a data availability section to the revised manuscript*

**1.7.    Specific comments and typographical errors**

All of the comments which relate to simple typographical errors have been fixed as per the reviewer's recommendation.

RC:    ***Line 11: "comparable (slower)" is contradictory. Which is it? -***

AR:    *We mean with this line to indicate that the values are comparable but slightly slower. We have changed this to "but slower" to improve clarity here.*

RC:    ***Lines 69-71: Please specify whether these trends in HCHO are for large urban areas, rural regions or across the country.***

AR:    *These values are for the whole country. This has been clarified in the text*

RC:    ***Line 186-187: "30 in the troposphere" is appropriate for a fixed tropopause, but doesn't the model have a dynamic tropopause that leads to variability in the number of levels in the troposphere?***

AR:    *The reviewer is right that GEOS-Chem has a dynamic tropopause. We have changed the text to:*

> *of which about 30 are normally below the dynamic tropopause*

RC:    ***Line 195: Is MIX a mosaic of regional inventories? As written this isn't clear.***

AR:    *The reviewer is correct that MIX is a mosaic of regional inventories. We have clarified that we are talking about the output from this mosaic in this sentence.*

RC:    ***Line 201: Provide the version of MEGAN that is used***

AR:    *We have added the version number (v2.1)*

RC:    ***Lines 206-207: Say where the drivers of MEGAN are from. Is LAI from the model fields or from MODIS?***

AR:    *We have added text to state that MEGANv2.1 uses MODIS LAI.*

RC:    ***Line 297: The meaning of the 11% is not clear. Is 11% the percent increase in vertical columns relative to slant columns after applying the AMF? Or is this an increase in the spatial correlation after applying the AMF?***

AR:    *This 11% value is derived from a comparison of the correlation coefficients. This is made clear in the revised text.*

RC:    ***Lines 347-348: Reiterate that the slant columns are used for 2008-2015.***

AR:    *This reiteration has been added as suggested.*

RC:    ***Line 352: Should "Median" be mean? That's what's shown in Figure 8.***

AR:    *The term "median" is used correctly here. While mean values are better for the graphics, for this particular statistic the median value is more useful as it is not perturbed by a handful of likely erroneous extremes.*

RC:    ***Line 384: Don't mature leaves emit more isoprene than young and senescing leaves (Guenther et al.,***

*2006)? This sentence needs to be revised to reflect that.*

AR: *This sentence was unclear in the original manuscript. We have replaced it with this sentence:*

> *Thermotolerance is one hypothesis that describes why leaves emit isoprene (Singaas et al., 1997). After a few weeks of emerging, the emission capacity of leaves peaks and subsequently declines with age.*

RC: *Lines 424-425: Why does the bottom-up emission inventory overestimate emissions from the transport sector when the inventory is for 2010 and the simulation year is 2014? Wouldn't this misrepresent the growth in this sector and so underestimate emissions from the transport sector? The reasoning for an overestimate is not immediately clear*

AR: *The expectation of a decrease in transport emissions 2010-2014 assumes transportation is emitting less VOC per unit as engines are modernized and regulations come into force.*

RC: *Line 456: Are the "sample mean" and "sample standard deviation" the regional mean and standard deviation of column HCHO for the whole country? If so, please clarify this in the text.*

AR: *We have clarified that the datasets here are for the whole country.*

RC: *Line 461: Rather than stating that further work is "outside the scope of this current study", more impact-ful to point out that this is an early demonstration of the efficacy of future geostationary satellites that will provide constraints on the temporal variability of HCHO.*

AR: *We have changed this to*

> *and is an early demonstration of how datasets with continuous measurements for a region (such as those from a geostationary satellite) could help constrain the temporal variability of HCHO*

**1.8. Comments on Figures and Tables**

RC: *Figure 1: ● City names (perhaps capital cities?) and the legend in the population map are not legible. ● The population map has a panel header ("India Population Map"). The other panels would benefit from headers too. ● The red outline could be made clearer in the top right map*

AR: *Figure 1A (the population map) has been replaced.*

Headers have been added to these figures.

RC: *Figure 2: ● Label "C" is missing. ● Consider choosing a colour bar with a more dynamic colour range. As is, JJAS emissions seem only marginally higher than emissions in other months. ● What are the isoprene emission units? "g C", "molecules C", or "mg C"?*

AR: *The missing label has been added. The color bar as been clarified as atoms C/cm2/s. As listed in Table 3, it is the MAM isoprene emissions which are generally higher rather than JJAS, and we feel the color scale on Figure 2 shows this adequately.*

RC: *Figure 3: Consider including initial NOx (or NO) concentrations for each isoprene HCHO yield line shown in the plot.*

AR: *We feel this information is better presented in a tabular format, the $NO_2$ ratios are present in Table 1. We have revised text in the manuscript to better highlight these values.*

RC: ***Figure 7:** • "Left" should be "top" and "right" should be "bottom" in the figure caption.*

AR: *The caption has been corrected.*

RC: ***Figure 9:** • Consider choosing a colour bar with a more dynamic colour range*

AR: *The choice of a color bar with maximum at 1ppb is deliberate, as this shows that for almost all of India the "High NOx" conditions are likely. Variations above 1ppb are less important than those around the 0.1ppb range.*

RC: ***Figure 12:** • Labels on the map are not legible. • Consider replacing "map imagery" with "road network".*

AR: *The caption has been changed as suggested. The underlying road network image has been changed to one that is clearer.*

RC: ***Table 1:** • Include a footnote in the table to indicate the source of variables (T, RH, BLH, O3, CO, NO2) provided.*

AR: *The footnote has been added as suggested*

**2. Reviewer #2**

**2.1. Major comment 1**

RC: ***Version 10-01 of the GEOS-Chem model is used in this analysis. The mechanism in v11-01 (released Feb 2017) includes large revisions on the isoprene oxidation mechanism effecting the HCHO yield. Specifically, the prompt yield of HCHO from the reaction of the isoprene peroxy radical with NO (RIO2 + NO) is increased by 24% (from 0.66 to 0.82). The revised mechanism is in good agreement with fully explicit mechanisms and generally reproduces observations (Marvin et al., 2017). Because we have more confidence in the updated mechanism, and because the mechanism is likely to affect the modeled HCHO columns shown here, the updated isoprene oxidation mechanism should be used in this analysis.***

AR: *The reviewer correctly notes that the isoprene oxidation mechanism has revised in the transition from GEOS-Chem v10-01 (used in this study) and v11-01 which was released in early 2017. The model calculations used in this project were initiated prior to the release of v11-01.*

We have recently installed GEOS-Chem v11-01. Re-doing the all calculations of this manuscript with a new model would not be viable in the normal turnaround time for a revised paper in ACP. Instead, to assess the likely impact of the difference in mechanism, we have compared model results for a test case month (April 2014).

For this test case, the model output was, in terms of HCHO columns, both qualitatively and quantitatively, similar for both versions of the model. There are no features in the model output for v11 that are not present in v10 (and vice versa). The model output vertical HCHO columns were strongly correlated without significant bias between them ([v11] = 0.96[v10] + $0.6 \times 10^{15}$, $r^2$ = 0.89). We decided not to include this in that manuscript.

**2.2. Major comment 2**

**RC:** *It is unclear here if the chemical mechanism used in the CABBA/MECA box model is the same as used in the GEOS-Chem simulation. Ideally, the same mechanisms would be used.*

**AR:** *The mechanisms are not the same. The mechanism used in GEOS-Chem is the NOx-Ox-HC-Aer-Br (a.k.a. "tropchem") mechanism which was in the standard release of GEOS-Chem v10-01. Text has been added directing the reader to Eastham et al (2014)* `(http://doi.org/10.1016/j.atmosenv.2014.02.001)` *which details the mechanism.*

We have clarified the box model mechanism is the full chemistry as described in Sander et al 2011 and the model's documentation, but without halogen, sulphur and mercury reactions. Additionally, the reader is directed to Taraborelli et al (2009) `(https://doi.org/10.5194/acp-9-2751-2009)` which discusses the "MIM2" isoprene mechanism used in the box model.

We note that, with respect to isoprene chemistry, both GEOS-Chem's chemical mechanism and MIM2 (used in the box model) are evaluated against the Master Chemical Mechanism (MCM) v2.1 in, Marvin et al 2017 and Tamborelli et al 2009 respectively and both generally performed well in reproducing the MCM results. We direct readers interested in this chemistry to these papers, and summarise that, while both comparisons are generally favourable when assessed against MCM, Marvin et al. (2017) finds that GEOS-Chem's mechanism slightly underestimates HCHO production in high-NOx conditions, whereas Tamborelli et al (2009) showed generally similar HCHO-results across the whole NOx range.

Please also note our response to Reviewer 1 above (1.5) that describes the text we have included.

**2.3. Major comment 3**

**RC:** *Recently, Zhu et. al (2016) found that the OMI SAO HCHO product used here was biased low by 37% compared to observations over the Southeastern United States. Do we have any reason to suspect that this bias may persist in other regions of the globe? I understand that in-situ HCHO observations are likely limited for this region; however, acknowledgement of the potential for bias and the need for validation is warranted.*

**AR:** *Unfortunately, it is not possible for us to evaluate whether a similar bias exists over India due to a lack of in situ data. To inform the reader about this potential bias in OMI we have added a reference to Zhu et al 2016 and a brief summary of their result. Specifically, we have included the following sentences:*

> *"Previous work using aircraft observations over Southeastern United States has shown that the OMHCHOv0003 OMI HCHO column data product has a low bias (37%), and that the GEOS-Chem model v9.02 had a 10% low bias (Zhu et al, 2016). In the absence of similar data over India, we cannot determine with certainty the generality of these biases, and have chosen to implicitly assume that the OMI data are not significantly impacted by systematic error.*

**2.4. Minor comments and typographical errors**

All of the comments which relate to simple typographical errors have been fixed as per the reviewer's recommendation.

**RC:** *Page 3, line 79: It is unclear what is meant by "this approach".*

**AR:** *This point has now been clarified. "Approach" has been changed to "relationship"*

**RC:** *Page 3, sentence starting at line 88: Wolfe et al. (2016) is cited as a reference for the uncertainty in HCHO production in the low NOx regime. In contrast, I think this paper suggests that the NOx dependency of HCHO production is well captured by updated chemical mechanisms. In any case, it is not clear to me why uncertainty in VOC emissions and in the low NOx yield means "both approaches provide useful insights". Perhaps it is more useful to point out that while the inverse model approach gives a more rigorous result, it comes at a higher computational expensive, and the local linear approach provides a useful approximation where there is little smearing.*

**AR:** *The reviewer is correct that one of the principal advantages of the local linear approach is its lower computational expense. We feel that a comparison of the relative merits of the available approaches would not improve the readability of the manuscript. We already provide a number of references in this paragraph that cover the different methods. This also includes the Marais reference that was missing in the submitted version of the paper.*

**RC:** *Page 7, line 201: List which version of MEGAN.*

**AR:** *We have included the version number (v2.1).*

**RC:** *Page 7, line 213: GEOS-Chem HCHO columns are averaged between 1300 and 1500 to provide comparison with OMI observations. Is this a hold-over from GEOS-Chem simulations run at coarse resolution with longer (60 min) timesteps? At the 10 minute chemistry timestep often used for 0.25 x 0.3125 simulations, I would have expected the output to be averaged over a smaller time frame (1300 to 1400). Perhaps the effect is small, but Is there a reason for the 1300 – 1500 averaging window? What are the chemistry and transport time steps used in this simulation?*

**AR:** *Due to the nature of the satellite orbit the local solar time at overpass can vary by over two hours, which is the reason for this wide averaging window. We found that due to the wide range of longitudes in a single swath, there is a range of approximately 100 minutes in local solar time for the pixels. This provides the motivation for adopting a two-hour window.*

In the high resolution run the transport timestep was 5 minutes and the chemistry timestep was 10 minutes. These values have been added to the text.

**RC:** *Page 8, lines 240-226. The box model analysis of isoprene oxidation as a function of NOx seems redundant with other recent publications, which also have the benefit of comparison to in-situ observations (Wolfe et al., 2016; Marvin et al., 2017). For example, figure 3 is reminiscent of Figure S3 in Marvin et al. 2017. At the very least, the box model results here should be compared with those in previous literature.*

**AR:** *Including the HCHO production from the oxidation of isoprene in Figure 3 was primarily intended to put into context the HCHO production from the oxidation of anthropogenic VOCs. With this figure we were updating a figure published in Palmer 2003 (`http://doi.org/10.1029/2002JD002153`) but in this current paper we have linked each calculation with the photochemical environment associated with each Indian forest. We do not wish to directly compare these results to those in dedicated studies of the HCHO yield–NOx relationship as these are qualitatively different exercises (in particular, our study adjusts many variables between model runs, not just NOx). Marvin et al (2017) has been added to the list of previous studies. We have added the following text at the end of the section:*

> The simulations reported in this work are intended to be specific to the Indian scenarios discussed.

**RC:** *Page 11, line 370: add a section refence (i.e. "as described in section 2.2")*

AR:    *This section reference has been added as per the reviewer's suggestion.*

**RC:    Page 15, lines 491: The procedure of deriving a posteriori emissions using the linear approach should be briefly described for readers unfamiliar with the process.**

AR:    *We have added the following text:*

> *We resolve seasonal a priori emissions of isoprene from observed HCHO columns by transposing the model linear relationship between isoprene emissions and HCHO columns.*

AR:    *Page 38, Table 3: Consider adding a "% change" column*

**RC:    This has been added as per the reviewer's suggestion.**

**3. List of changes**

*The following is a list of all the changes made in the document. Line numbers refer to the positions in the manuscript appended to this document with highlighted changes*

- *Line 3: "spatial" changed to "horizontal"*

- *Line 8: "neighbors" changed to "neighbour"*

- *Line 11: Added "but" before "slower"*

- *Line 12 and 22: Minor changes regarding hyphens*

- *Line 22: removed "but there are large year to year variations"*

- *Line 24: removed "only"*

- *Line 25: changed "during March to April" to "only during March and April"*

- *Line 66 and 108: contracted "Ozone Monitoring Instrument" to "OMI"*

- *Line 71: added "nationwide"*

- *Line 75: added "by"*

- *Line 77: added "these" before "data"*

- *Line 80: changed "approach" to "relationship". Please noted that the highlighting that suggests references have been removed from this line is due to a minor bug and is incorrect.*

- *Line 86: Added text referring to Marais et al (2012).*

- *Line 90: changed "NO" to "NO$_x$"*

- *Line 140: changed "detailed" to "detail"*

- *Line 151: added paragraph referring to work of Zhu et al (2016).*

- *Line 164: added paragraph regarding model mechanisms.*

- *Line 176: added "but without halogen, sulphur, and mercury chemistry"*

- *Line 180: corrected citation format.*

- *Line 203: changed "signal-levels" to "sigma-levels"*

- *Line 204: changed to "of which about 30 are typically below the dynamic tropopause"*

- *Line 205: added sentence regarding mechanism*

- *Line 211: added sentence discussing to inventories. This replaces text from Line 230 which has been removed.*

- *Line 214: added "mosaic"*

- *Line 215: removed repeated word*

- *Line 222: added version number*

- *Line 236: removed "Due to computational limitations..." sentence.*

- *Line 272: changed "(>1 ppbv)" to "(averages generally around or above 1 ppbv)"*

- *Line 275: changed "getting oxidised" to "oxidation"*

- *Line 282: added "(c.f. $NO_2$ mixing ratios in Table 1 and yields reported in Table 2)"*

- *Line 284: added "The simulations reported in this work are specific to the Indian scenarios discussed."*

- *Line 290: added "from a comparison of the correlation coefficients"*

- *Line 315: changed "North" to "north"*

- *Line 320: changed "more of the spatial distribution" to "more of the model spatial distribution than the observed slant columns"*

- *Line 371: added "slant" before "columns"*

- *Line 372: changed "those" to "observed distributions"*

- *Line 375: removed "that"*

- *Line 376: changed "of" to "from"*

- *Line 379: Added sentence referring to Mahajan et al. (2015)*

- *Line 396: changed "above" to a section reference*

- *Line 409: re-wrote sentence regarding leaf thermotolerance*

- *Line 413: added hyphen*

- *Line 432: spelling correction*

- *Line 452: changed "like" to "likely"*

- *Line 457: spelling correction*

- *Line 463: used order symbol*

- *Line 470: changed "The transportation sector" to "Emissions from the transportation sector"*

- *Line 471: changed "has peak" to "peak"*

- *Line 477: changed "data collected using" to "data collected from"*

- *Line 482: added "for the whole country"*

- *Line 490: revised statement regarding further work*

- *Line 503: changed "that correspond to MODIS land cover" to "with MODIS land cover"*

- *Line 508: corrected citation format*

- *Line 514: revised statement regarding linear relationship*

- *Line 528: added sentence regarding Zhu et al 2016*

- *Line 558: spelling correction*

- *Line 559: changed "VOC" to "VOCs"*

- *Line 563: moved "only" within this sentence*

- *Line 566: changed "proposed an argument" to "argue"*

- *Line 585: changed "Achieving this capability is greatly enhanced" to "Our ability to achieve this capability is improved"*

- *Line 586: added "aboard Sentinel-5P"*

- *Line 589: added Data availability section*

- *Figure 1: Figure 1A replaced with clearer version and caption edited accordingly. Figure 1B slightly edited to give greater contrast. Header labels added to Figures 1B and 1C. Two grammar corrections in caption/*

- *Figure 2: Units clarified as atom C/cm$^2$/s. Label added to 2C.*

- *Figure 3: In caption, changed "per unit C" to "per-C"*

- *Figure 4: Himalayas and Rann of Kutch highlighted*

- *Figure 5: "Northeast", "East forest", "Kerala" labelled*

- *Figure 7: Caption corrected*

- *Figure 12: Underlying imagery changed, and caption accreditation changed accordingly*

- *Table 1: Caption changed to indicate T, RH and BLH are from GEOS-Chem*

- *Table 2: changed "per unit C" to "per-C"*

- *Table 3: Units clarified as atom C/cm$^2$/s. % change from a priori column added*

**4. Final manuscript with highlighted changes**

*The revised manuscript it appended to this document with changes highlighted. Removed text is shown in red, inserted text in blue.*

[revised manuscript text omitted]